# Transferring disentangled representations: bridging the gap between synthetic and real images

**Jacopo Dapueto    Nicoletta Noceti    Francesca Odone**
jacopo.dapueto@edu.unige.it
{nicoletta.noceti,francesca.odone}@unige.it
MaLGa-DIBRIS, Università degli studi di Genova, Genova, Italy

## Abstract

Developing meaningful and efficient representations that separate the fundamental structure of the data generation mechanism is crucial in representation learning. However, Disentangled Representation Learning has not fully shown its potential on real images, because of correlated generative factors, their resolution and limited access to ground truth labels. Specifically on the latter, we investigate the possibility of leveraging synthetic data to learn general-purpose disentangled representations applicable to real data, discussing the effect of fine-tuning and what properties of disentanglement are preserved after the transfer. We provide an extensive empirical study to address these issues. In addition, we propose a new *interpretable* intervention-based metric, to measure the quality of factors encoding in the representation. Our results indicate that some level of disentanglement, transferring a representation from synthetic to real data, is possible and effective.

## 1   Introduction

Developing meaningful, reusable and efficient representations is a critical step in representation learning [1, 58, 57, 67]. Disentangled Representation Learning (DRL) [1, 40, 24, 67] aims to learn models that can identify and disentangle underlying Factors of Variation (FoVs), hidden in the observable data. These models encode them in an interpretable and compact shape [31, 9, 1, 73], independently from the task at hand [22, 39, 66, 67]. Moreover, DRL enhances explainability, robustness, and generalization capacity across various applications [67]. Disentangled representations have been shown useful for various downstream tasks, such as FoVs prediction [41, 40], image generation [72, 48, 43, 42, 59] and translation [21, 19, 38], fair classification [56, 39], abstract reasoning [66, 63], domain adaptation [35], and out-of-distribution (OOD) generalization [11, 20].

While all the abovementioned methods may rely on different definitions of disentanglement (see just as examples [1, 23, 64]), and in this sense a comprehensive comparison is hard, they usually share the observation that some level of supervision on the FoVs is beneficial for disentanglement. However, labelling every single factor to achieve fully supervised disentanglement is costly or even unfeasible [70, 52]. For this reason, DRL has been mostly validated on synthetic or simulated data, usually acquired on purpose [11, 41, 60], and there is a limited understanding of the potential of DRL to address general-purpose representation tasks, as well as the specific challenges of the real world (e.g. the presence of clutter and occlusion, correlation between factors [11], etc.). Such challenges may prevent the model from learning perfectly disentangled representations [65].

In this work, we propose the adoption of Disentangled Representation (DR) transfer to deal with complex realistic/real dataset. DL transferring was explored in [20], where Source models learnt in an unsupervised manner were transferred to a Target dataset, by transferring hyperparameters. The authors observed a limited effectiveness in the direct transfer of representations. Instead, Dittadi et al. [11] found out that disentangled representation can help in OOD generalization from a simulated to a smaller real dataset. In both cases, the study involved very specific types of dataset, built to emulate

the real one in every detail. Recently, Fumero et al. [18] addressed disentanglement in real data without the need for FoVs annotation, leveraging the knowledge extracted from a diversified set of supervised tasks to learn a common disentangled representation to be transferred to real settings. We follow a different direction, setting up a very straightforward and generalizable procedure: we resort to a weakly supervised approach [41, 25, 26] to learn DRs on Source datasets where the FoVs are known and annotated, to then transfer (with no supervision) such representation to a Target dataset where the FoVs are not known or available. Our final aim is to consider real datasets as a Target, while synthetic data (where FoVs annotation is easy to obtain) can be employed as a Source.

The paper presents three main contributions: **(1) a novel metric** to assess the quality of disentanglement, which is *interpretable*, classifier-free and informative on the structure of the latent representation; **(2) a DR transfer methodology** to Target datasets without FoV annotation; **(3) an extensive experimental analysis** that considers different (Source, Target) pairs and quantitatively assesses the expressiveness of the learnt DR on Target of different nature (including the case where the gap between Source and Target is large), taking into consideration the main expected properties of disentangled representation. We discuss the role of fine-tuning and the need to reason on the distance between Source and Target datasets.

The paper is organized as follows. In Section 2, we propose and discuss our new intervention-based metric, OMES. In Section 3.3, we introduce our transfer approach to DRL, and provide a thorough analysis of different types of transfer scenarios (synthetic to synthetic, synthetic to real, real to real). Section 5 is left to the conclusions.

## 2 Evaluating the quality of disentanglement

### 2.1 Background

While there is no universally accepted definition of disentanglement, there is common agreement on the properties that a DR should have [12, 54, 66, 1, 57, 2]:

**Modularity** [55]: A factor influences only a portion of the representation space, and only this factor influences this subspace. This is achievable if the FoV are independent, meaning that a variation in one FoV does not affect others.

**Compactness** [55]: The subset of the representation space affected by a FoV should be as small as possible (ideally, only one dimension). This property is also called *completeness* in [13].

**Explicitness** [54]: DR should explicitly describe the factors, thus it should favour FoVs classification.

The taxonomy presented in [12] groups all metrics in three families (see a summary in Table 6 in App.): **Intervention-based** metrics compare codes by intervention, either creating subsets of data in which one or more factors are kept constant (*BetaVAE* [24] and *FactorVAE* [27]), or in which only one factor is varying (*RF-VAE* [28]), and predicting which factors were involved in the intervention; **Predictor-based** metrics use regressors or classifiers to predict factors from DR (*DCI Disentanglement* [13] and *SAP* [32]) or intervened subsets (*BetaVAE*, *FactorVAE* and *RF-VAE*); **Information-based** metrics leverage information theory principles, such as mutual information, to quantify factor-DR relationships (*Mutual Information Gap (MIG)* [8, 12], *MED* [6], *Modularity* [55] and *InfoMEC* [25]).

Intervention-based metrics have the advantage of providing control over the factor and the corresponding representation. However, they are all based on classifiers, thus they depend on method, hyperparameter settings and model capacity. The latter consideration can be extended to all Predictor-based metrics. On the other hand, Information-based methods are mainly ground on the computation of Mutual Information, which is dependent on an estimator and its parameters [51, 7].

Motivated by these limitations, we introduce in the next section a new metric, to the best of our knowledge, the *first* classifier-free intervention-based metric.

### 2.2 Our metric: OMES

OMES *(Overlap Multiple Encoding Scores)* is an intervention-based metric measuring the quality of factor encoding in the representation while providing information about its structure: we measure *modularity*, analyzing how the FoVs overlap, and *compactness*, detecting and quantifying how a factor is encoded in the dimensions of the representation.

**Algorithm 1** Compute association matrix $S$ between dimensions and FoVs

---

**Require:** $D_\Phi = [\boldsymbol{R}^1, \boldsymbol{R}^2, \boldsymbol{k}], n$             ▷ $n$ number of FoVs
**Ensure:** $\boldsymbol{R}^1, \boldsymbol{R}^2 \in \mathbb{R}^{N \times m}, \boldsymbol{k} \in \mathbb{R}^N$      ▷ $m = |\mathcal{A}|$, $N$ number of pairs in $D$
 1:  $S \leftarrow \text{ZEROS}(m, n)$
 2: **for** $j = 1$ to $n$ **do**
 3:    $\boldsymbol{R}^1_j = \boldsymbol{R}^1[\boldsymbol{k} == j, :]$ and $\boldsymbol{R}^2_j = \boldsymbol{R}^2[\boldsymbol{k} == j, :]$
 4:    **for** $h = 1$ to $m$ **do**
 5:       $PC = PearsonCorr(\boldsymbol{R}^1_j[:, h], \boldsymbol{R}^2_j[:, h])$
 6:       $S[h, j] = 1 - abs(PC)$
 7:    **end for**
 8: **end for**
 9: **return** $S$                               ▷ $m \times n$ matrix

---

**Algorithm 2** Overlap score of FoV $j$: **OS**

---

**Require:** matrix $S$, FoV index $j$
**Ensure:** $S \in \mathbb{R}^{m \times n}$
 1: scores $\leftarrow \text{ZEROS}(m)$
 2: **for** $h = 1$ to $m$ **do**    ▷ dimension $h$
 3:   $i_S \leftarrow \text{ZEROS}(n); i_S[j] = 1$
 4:   scores$[h] = 1 -\text{MAE}(i_S, S[h, :])$
 5: **end for**
 6: **return** POOLING(scores)

---

**Algorithm 3** Encoding score of FoV $j$: **MES**

---

**Require:** matrix $S$, FoV index $j$
**Ensure:** $S \in \mathbb{R}^{m \times n}$
 1: scores $\leftarrow \text{ZEROS}(m)$
 2: **for** $h = 1$ to $m$ **do**    ▷ dimension $h$
 3:   $i_S \leftarrow \text{ZEROS}(m); i_S[h] = 1$
 4:   scores$[h] = 1 -\text{MAE}(i_S, S[:, j])$
 5: **end for**
 6: **return** POOLING(scores)

---

Given an image $X$, with $\Phi$ its mapping into a $d-$dimensional latent disentangled space, $\Phi(X) = r$, $r \in \mathbb{R}^d$. We discard dimensions whose empirical standard deviation is extremely small ($< 0.05$), meaning that the dimensions are inactive [68, 10].This leaves us with a subset of $m \leq d$ active dimensions, to which we will refer in the following.

Let $D$ be a dataset formed by image pairs, $D = \{(X_i^1, X_i^2, k_i)\}_{i=1}^N$, where $X_i^1, X_i^2$ are two images that differ for only the FoV $k_i$.

OMES requires computing a weighted *association* matrix $S$ between the dimensions of the representation and the FoVs, with higher association values if the factor is encoded in a certain dimension (see Algorithm 1): we consider the representations of the image pairs in dataset $D$, obtaining $D_\Phi$. In matrix notation we may write it as $D_\Phi = [\boldsymbol{R}^1, \boldsymbol{R}^2, \boldsymbol{k}]$, where the pair $\boldsymbol{R}^1$ and $\boldsymbol{R}^2$ are $N \times m$-dimensional matrices with each row $i$ is the representation of the $i$-th image pair $\Phi(X_i^1) = r_i^1$ and $\Phi(X_i^2) = r_i^2$ respectively. For each FoV $k$ we extract the rows of $D_\Phi$ such that the $i$-th entry of vector $\boldsymbol{k}$ is $k_i = k$, we call this set $D_\Phi^k$. Each entry $S[h, j]$ of the association matrix relates a dimension $h$ of the estimated disentangled representation with a FoV $j$. Its value is in the range $[0, 1]$ with elements close to 1 corresponding to a dimension that effectively captures the variations of a FoV. The association is based on a correlation analysis: since the samples from $D_\Phi^k$ are paired to differ only for the FoV $k$, we expect a good representation *not to correlate* where such FoV is encoded. To ease interpretability, we transform the obtained values (see Algorithm 1, line 6) so that high values denote a strong association between dimension and FoV.

When the model exhibits perfect disentanglement, each row and column of the association matrix $S$ present just one element with a high association, corresponding to the only dimension where the factor is encoded. We thus measure the level of disentanglement through similarity with an ideal array, where the association matrix shows all 0s but in the positions of the correct associations, where there are 1s.

We rely on the above considerations to derive our metric as a linear combination of two main contributions. The *Overlap Score* (OS) penalizes the overlap of different FoVs in the same dimensions (Algorithm 2 — in this case, each row of $S$, associated with a dimension, is compared with the ideal array) and hence measures Modularity, while the *Multiple Encoding Score* (MES) penalizes the encoding of the same factor into different dimensions (Algorithm 3 — in this case each column

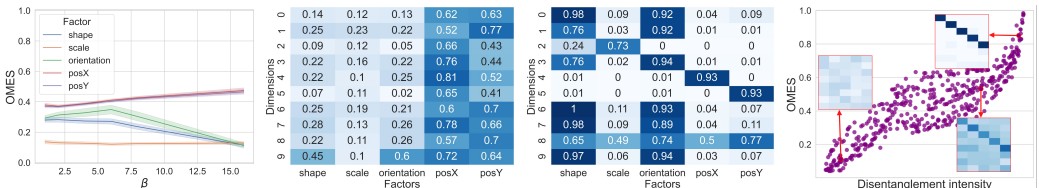

Figure 1: Dataset *Noisy-dSprites*: **Left:** Scores of the proposed metric for each FoV, $\alpha$ is fixed to 0.5. **Center Left:** Association matrix of an unsupervised model ($\beta = 6$). **Center Right:** Association matrix of a weakly-supervised model. **Right:** Scores of synthetic Association matrices simulating underfitting, partial disentanglement and almost perfect disentanglement.

of $S$ corresponding to a FoV is compared with the ideal array) measuring Compactness. In both algorithms, we derive a vector summarizing the contribution of all dimensions for the FoV.

The final score in $[0, 1]$ (higher values meaning higher disentanglement) can be obtained with a pooling (either MAX or AVERAGE) on the vector. The OMES metric is computed as

$$OMES(S) = \frac{1}{n} \sum_{j=1}^{n} \alpha \, \text{OS}(S, j) + (1 - \alpha) \, \text{MES}(S, j). \tag{1}$$

With $\alpha = 0$, OMES only measures the Compactness of the representation (MES component); with $\alpha = 1$, instead, our metric measures the Modularity only (with OS). Values of $\alpha$ in the interval $(0, 1)$ can be used to balance the importance of both contributions.

**Relation with existing metrics**. To the best of our knowledge, the only metrics capturing more than one property are DCI [13] and the very recent InfoMEC [25]. Differently from DCI, our metric is intervention-based with no influence on the choice of the specific classifier that may inevitably impact the results, as observed in [7]. With respect to InfoMEC, that must be applied to quantized latent codes, our metric is more general and accepts continuous latents.

OMES is based on the intervention of the FoVs, thus we require the FoV to be (at least partially) known: in particular, samples are coupled so that they differ in one FoV only. In this, OMES differs from existing intervention-based metrics [24, 27] in which the intervention is the opposite (samples have only one FoV in common). Our pairing requires less supervision, and it is usually easier to obtain during data acquisition (for instance, from videos [41]). In addition, it has been shown that this type of pairing provides more guarantees on disentanglement properties [60, 41].

Finally, compared to Information-based methods, we exploit Correlation instead of Mutual Information, hence we do not need its estimation that can be sensitive to parameters choice (e.g. granularity of the discretization [7]) and choice of estimator [51, 7].

### 2.3 OMES assessment

We now analyze OMES, extending previous studies on the unsupervised [40] and weakly supervised [41] setting. As for the *unsupervised case*, we exploit available 5400 trained models from [40] (3 datasets, 6 values for $\beta$, 50 random seeds, 6 unsupervised methods:$\beta$-VAE [24], FactorVAE [27], $\beta$-TCVAE [8], DIP-VAE-I [32], DIP-VAE-II[32], AnnealedVAE [5]); in this section we report an analysis on Noisy-dSprites, the remaining 2 benchmarks can be found in the Appendix B.1 and B.2. Instead, for the *weakly supervised case* trained models are not available, so we reproduce the models as in [41] training them on Shapes3D and on other datasets that can be found in Appendix B.3.

**OMES interpretation.** The metric, by construction, allows us to compute the overall score and a score for each FoV separately: we can thus interpret the effect of hyperparameters on the single FoV, and evaluate the FoV separately in each dimension of the representation. Moreover, by inspecting the metric at a factor level, we may identify uneven behaviours (e.g. models performing similarly on average but for different contributions from the factors).

Fig. 1 (Left) shows the metric scores for different values of $\beta$ keeping the different FoV separated: the FoVs less affected by reconstruction (e.g. `PosX` and `PosY`) exhibit an increasing disentanglement score as $\beta$ grows. On the other hand, `Shape` and `Orientation` present a maximum value around $\beta = 6$ and then decrease because they are more susceptible to the reconstruction quality, which degrades for larger values of $\beta$. In Fig. 1 (Center Left) an association matrix $S$ is generated from one

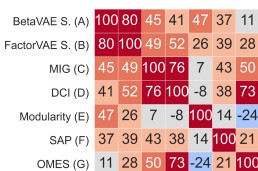 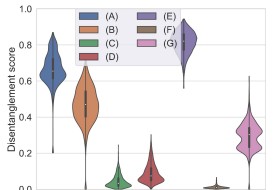 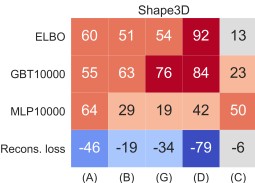

Figure 2: **Left:** Rank-correlation between metrics of models trained on *Noisy-dSprites*. **Center:** Scores distribution of the metrics on Noisy-dSprites. **Right:** Rank correlations (Spearman) of ELBO, reconstruction loss, and the test accuracy of a GBT and a MLP classifier trained on 10,000 labelled data points with disentanglement metrics. In all plots OMES is computed with $\alpha = 0.5$.

of the unsupervised models ($\beta$-VAE) trained with $\beta = 6$: `Shape` and `Orientation` are encoded in the same dimension (overlapping) and produce lower values because of the reconstruction, while `PosX` and `PosY` are encoded in multiple dimensions and mostly overlapping. `Scale` does not seem to be well represented. In Fig. 1 (Center Right) we report the association matrix obtained by a weakly-supervised model (Ada-GVAE): the factors `PosX`, `PosY` and `Scale` are disentangled while `Shape` and `Orientation` are encoded in the representation with high intensity in different dimensions, with overlaps. Fig.1 (Right) shows OMES values ($\alpha = 0.5$) computed over synthetic association matrices $S$, obtained by perturbing the ideal (diagonal) one. The perturbation aims to simulate 3 scenarios: noisy matrices where the disentanglement can be more or less strongly derived; models exhibiting partial disentanglement; models with no disentanglement. As it can be appreciated, the metric score nicely reflects the disentanglement intensity.

**Agreement of OMES with other disentanglement metrics.** We extend the analysis in [40]. The rank-correlation (e.g. Fig. 2 (Left)) between the previously proposed metrics and our OMES (for $\alpha = 0.5$) shows that the latter has a high level of correlation with MIG and DCI, but mild correlation with BetaVAE, FactorVAE (OMES is based on the opposite intervention type), and Modularity. This is consistent on all benchmarks. We show in Fig. 2 (Center) the score distribution of the metrics, computed on the whole set of models. We observe OMES produces a wider range of values with respect to MIG and DCI: our metric looks more descriptive, similarly to BetaVAE and FactorVAE.

**Agreement with performance metrics.** Similarly to [41], we consider the more informative weakly-supervised setting and discuss the rank correlation between our metric and performance evaluations (ELBO, reconstruction loss, and test error of FoVs classifier). Our analysis, reported in Fig. 2 (right) shows that OMES performs similarly to DCI, negatively correlated with the Reconstruction loss, and positively with the ELBO. It also correlates with the performances of GBT10000 (the classifier we will use in the experiments) while it mildly does with MLP10000. This empirical evidence is in line with what was observed in [11]. *The correlation with the classification score is a sign that OMES is able to capture the property of expliciteness of the representation, although it is not directly measured by our metric.* It is worth mentioning the correlation of OMES with the performance metrics is more stable than what is obtained by other metrics across different datasets (see the Appendix B.3).

## 3 Transferring disentangled representations

Fully unsupervised disentangled representation learning has been shown unsatisfactory in many scenarios [40]. However, annotating the FoVs can be a very critical and uncertain process. In this section, we propose a general-purpose methodology for transferring disentangled representations learned from supervised synthetic or simulated data to an unsupervised dataset (in terms of the FoVs). This approach allows us to evaluate the effectiveness of disentangled representations transfer, and its potential in real-world applications.

### 3.1 Our methodology and research questions

Most of the focus in learning disentangled representations has been on synthetic datasets whose ground truth factors exhibit perfect independence by design [44, 53, 15, 34, 4]. Instead, real-world

Table 1: Summary of the datasets and their properties. * in the $\#FoV$ refers to the possible presence of hidden factors.

| Dataset | Real | 3D | Occlusions | #FoV | Independence | Complete annotation | Resolution | #Images |
|---------|------|-----|------------|------|--------------|---------------------|------------|---------|
| dSprites | ✗ | ✗ | ✗ | 5 | ✓ | ✓ | $64 \times 64$ | 737K |
| Noisy-dSprites | ✗ | ✗ | ✗ | 5 | ✓ | ✓ | $64 \times 64$ | 737K |
| Color-dSprites | ✗ | ✗ | ✗ | 6 | ✓ | ✓ | $64 \times 64$ | 4,4M |
| Noisy-Color-dSprites | ✗ | ✗ | ✗ | 6 | ✓ | ✓ | $64 \times 64$ | 4,4M |
| Shapes3D | ✗ | ✓ | ✓ | 6 | ✓ | ✓ | $64 \times 64$ | 480K |
| Isaac3D | ✗ | ✓ | ✓ | 9 | ✓ | ✓ | $128 \times 128$ | 737K |
| Coil100-Augmented | ✓ | ✓ | ✓ | 4 | ✓ | ✓ | $128 \times 128$ | 1,1M |
| RGB-D Objects | ✓ | ✓ | ✓ | 3* | ✗ | ✗ | $256 \times 256$ | 35K |

scenarios present several challenges that we want to investigate in our analysis.

We consider $\beta$-VAE models with weakly supervised learning specifically we adopted Ada-GVAE [41], for its simplicity and its sampling strategy similar to our metric, on a *Source* Dataset, using pairs of images that differ in $\ell$ factors of variation. We set $\ell = 1$ as it was shown to lead to higher disentanglement [41]. Following [11], we vary the parameter $\beta$ in $\{1, 2\}$, sufficient to achieve high disentanglement with weak supervision [11, 41].

We evaluate the quality of the disentanglement in a transfer learning scenario, assessing the transferability of the disentangled representation on a *Target* Dataset, with the final aim of targeting real scenarios. The evaluation we report considers our metric OMES, as well as DCI and MIG, the most widely used metrics in the literature [50, 18, 17, 14, 69, 25, 45]. Moreover, in accordance to [40, 41, 11], we evaluate the quality of the disentanglement also in terms of accuracy w.r.t. a downstream classification task, with a classifier per FoV. We evaluate the latter in two modalities: (1) Considering the entire representation and (2) Selecting with OMES the dimension of the representation that best encodes the FoVs. Our analysis addresses three main research questions:

**Q1 -** *How well does disentanglement transfer, and how much does it depend on the distance between Source and Target Dataset?*

We will consider different transfer learning scenarios (syn2syn, syn2real, real2real) and pairs (*Source*, *Target*) datasets with different distances.

**Q2 -** *Which properties of a DR are preserved on the Target Dataset?*

We will discuss *Explicitness* of the representation (through FoV classification), *Compactness* (analysing the component MES of OMES, the MIG metric, as well as the performances of the one-dimensional representation), *Modularity* (relying on the OS component of OMES and on DCI).

**Q3 -** *How effective is fine-tuning on the disentanglement?*

We will consider the performances of the FoVs classification, the compactness and the modularity on the *Target dataset* before and after fine-tuning.

## 3.2 Datasets

In our analysis, we consider both synthetic and real datasets offering different challenges, a summary of their properties is in Tab. 1. Some of the datasets are *DRL-compliant*, meaning that there is full independence between the FoVs (this is reported in column *Indepencence*), and FoVs appear in all their possible combinations. This is easy to achieve if the dataset is specifically tailored for DRL, but it can not be easily obtained in general.

**dSprites**[44] is a dataset of 2D shapes generated from 5 ground truth FoVs: `Shape`, `Scale`, `Rotation`, `x` and `y` `Positions`. Variants of the dataset have been proposed: in **Noisy-dSprites** the background is filled with uniform noise; **Color-dSprites** includes `Color` as an additional FoV; **Noisy-Color-dSprites** adds uniform noise to the latter. We refer to them as: **N-dSprites**, **C-dSprites** and **N-C-dSprites**.

**Shapes3D** [4] is a dataset of 3D shapes, generated from 6 ground truth FoVs: `Floor colour`, `Wall colour`, `Object colour`, `Scale`, `Shape` and `Orientation`. It is characterized by the presence of *Occlusions*.

**Isaac3D** [47] is a synthetic dataset of a 3D scene of a kitchen where a robot arm is holding objects in a variety of configurations. It is characterized by 9 *real-world complex* FoVs, including `robot movements`, `camera height`, `environmental conditions` (e.g. lighting).

There are few real datasets available specifically meant for DRL. [20] is a collection of datasets covering the transitions from simulated to real data, which is, however, not fully available at the moment. [11] is not appropriate for our analysis since the real data section is very small compared to the complexity of the task. We consider instead real benchmarks proposed for classification tasks, chosen to reflect some of the real-world challenges but possessing some "semantic connection" with the synthetic dataset we refer to, e.g. in terms of the expected FoVs. This allows us to reason on the potential of transferability. Example images are in Appendix C.1.

**Coil** is derived from Coil100 [46]. The original dataset contains 7200 real color images of 100 objects. The objects were placed on a motorized turntable against a black background. The turntable was rotated to vary object pose w.r.t. a fixed camera, producing self-occlusions and 2D silhouette changes. We augment the original dataset with two additional FoV, a planar rotation (9 angles) and a scaling (18 values). Therefore, we identify 4 FoV (`Objects`, `Pose`, `Rotation` and `Scale`) that, by construction, are independent. To consider in our analysis a real dataset visually related to dSprite, we derived a binary version of Coil, called **Coil(bin)**, by applying Otsu's thresholding [49].
**RGBD-Objects** [33] is a dataset of 300 common household objects acquired by a RGB-D camera. The objects are organized into 51 `Categories` and a varying number of *instances* for each category. For each object, 3 video sequences have been acquired with different camera heights (`Elevation`) so that the object is viewed from different angles while rotating (`Pose`). Then, images have been cropped so that the object is always in a central position. For our experiments, we used a subset with one object instance per category to make it semantically similar to Coil100 but with the additional complexity of variability in the background, presence of occlusions and clutter. Hence, we control 3 FoVs (`Category`, `Elevation`, `Pose`), but other factors are *hidden or not annotated* (e.g. Background, Illumination, etc.) due to a realistic acquisition protocol. We refer to RGBD-Objects as **RGBD**. We also use a variant of the dataset, including depth maps only, referred to as **RGBD(depth)**.

### 3.3   Experimental analysis

**Implementation details.** We trained 20 different models (10 random seeds $\times$ 2 values of $\beta$) for each *Source* dataset. We adopted the same training strategy as in [11] (see Appendix C.2). As for FoVs classification, following [11, 41], we consider Gradient Boosted Trees (GBT) [16] and a Multilayer Perceptron (MLP) [37] with 2 hidden layers of size 256. Since the specific choice of a classifier is not crucial for our analysis, here we report GBT, MLP can be found in Appendix C.7. Fine-tuning to the *Target* dataset of the VAE models is unsupervised and it is carried out for 50k steps.

**Tables description.** The tables group different experiments based on the *Target* dataset. For each FoV, we report under the name the number of values the factor can assume (i.e. its *granularity*). The tables report the average classification performance over the 20 models, before and after fine-tuning. The latter is reported in parenthesis in terms of gain or loss w.r.t. the performance before the fine-tuning. *All* is the average performance of all FoVs.
The column *Pruned* highlights the two different representation modalities: if the classifier is trained on the whole representation (✗), or using only one dimension, i.e. the one showing the strongest encoding of a certain FoV according to the OMES metric (✓). As already mentioned, a good performance of the former is an indication of explicitness, while the latter is a positive sign of compactness. Tables also report metrics assessing Modularity (our MES and DCI) and Compactness (our OS and MIG).
Note that we exploit the interpretability of OMES in the transfer learning process to select the most representative dimension of the representation for the classification (the "Pruned" columns).

**(1) Synthetic to synthetic.** As a baseline, we consider the case in which both *Source* and *Target* datasets are synthetic and we have access to the annotation of the FoVs, they are DRL-compliant. If Source and Target have the same FoVs (S=dSprites with T=Noisy-dSprites or S=Color-dSprites with T=Noisy-Color-dSprites, see Table 2) we observe that pruning the representation to just one dimension maintains, on average, stable performances. This shows that the *compactness* of the representation is preserved for the Target dataset, both before and after fine-tuning.
Fine-tuning allows for improved performance in terms of *explicitness* preserving the remaining properties of the representation, also in the case of the pruned representation. The `Orientation` FoV is difficult in these datasets as it suffers from reconstruction errors. We increase complexity by adding a new FoV to the Target dataset (S=dSprite with T=Color-dSprite, see Table 2).

Table 2: Quantitative evaluation of transferred disentangled models using the dSprites family of datasets. We transfer from a Source (ST) to a Target Dataset (TD). We report the average classification accuracy obtained with GBT on the full and the pruned representations (see text). The last columns on the right report a comparison between disentanglement metrics, including MES and OS.

| SD | TD | Pruned | Mean accuracy on FoVs(%) | | | | | | | Modularity(%) | | Compactness(%) | |
| | | | Color (7) | Shape (3) | Scale (6) | Orientation (40) | PosX (32) | PosY (32) | All | Our (OS) | DCI | Our (MES) | MIG |
|---|---|---|---|---|---|---|---|---|---|---|---|---|---|
| dSprites | N-dSprites | ✗ | | 61.1 (+11.2) | 47.1 (+12.0) | 6.7 (+7.5) | 17.8 (+27.9) | 17.1 (+28.1) | 30.0 (+17.3) | 31.8 (+6.3) | 22.0 (+4.0) | 32.1 (+3.9) | 13.9 (+6.9) |
| | | ✓ | | 52.1 (+6.6) | 44.7 (+8.3) | 3.8 (+3.5) | 14.3 (+22.5) | 14.3 (+21.9) | 25.8 (+12.6) | | | | |
| | C-dSprites | ✗ | 30.8 (+35.4) | 94.2 (-1.9) | 86.9 (+0.5) | 44.8 (-4.2) | 76.3 (-5.1) | 75.7 (-5.3) | 68.1 (+3.2) | 61.0 (+1.6) | 42.9 (+10.4) | 72.3 (+3.5) | 34.0 (+0.3) |
| | | ✓ | 26.2 (+6.8) | 76.7 (+2.6) | 77.0 (+2.4) | 17.1 (+1.0) | 74.9 (-4.1) | 74.6 (-4.3) | 57.8 (+0.7) | | | | |
| C-dSprites | N-C-dSprites | ✗ | 33.0 (+65.2) | 41.6 (+7.3) | 28.6 (+16.8) | 2.9 (+0.9) | 9.0 (+19.7) | 9.40 (+20.2) | 20.7 (+21.7) | 28.9 (+0.3) | 5.4 (+2.2) | 29.0 (-1.9) | 1.8 (+1.0) |
| | | ✓ | 29.1 (+52.3) | 39.3 (+4.5) | 25.3 (+11.2) | 2.6 (+0.4) | 6.0 (+7.1) | 5.3 (+7.4) | 17.9 (+13.8) | | | | |

Table 3: Target dataset: Shapes3D (see Table 2).

| SD | TD | Pruned | Mean accuracy on FoVs(%) | | | | | | | Modularity(%) | | Compactness(%) | |
| | | | Floor Hue (10) | Wall Hue (10) | Object Hue (10) | Scale (8) | Shape (4) | Orientation (15) | All | Our (OS) | DCI | Our (MES) | MIG |
|---|---|---|---|---|---|---|---|---|---|---|---|---|---|
| dSprites | Shapes3D | ✗ | 78.0 (+14.1) | 80.3 (+13.1) | 43.6 (+25.6) | 25.4 (+9.6) | 55.5 (+18.6) | 35.3 (+0.8) | 53.0 (+13.6) | 26.9 (+0.6) | 9.6 (+9.0) | 23.9 (+1.1) | 5.3 (-0.4) |
| | | ✓ | 63.5 (-2.0) | 59.5 (+4.2) | 28.5 (+13.7) | 21.3 (+4.4) | 43.6 (+11.4) | 23.4 (-0.8) | 40.0 (+5.1) | | | | |
| C-dSprites | Shapes3D | ✗ | 82.1 (+8.0) | 79.4 (+12.3) | 46.8 (+28.8) | 30.1 (+10.4) | 53.7 (+32.6) | 39.8 (+5.1) | 55.3 (+16.2) | 30.6 (+0.8) | 13.7 (+7.4) | 28.2 (+2.1) | 8.4 (-1.4) |
| | | ✓ | 60.6 (-5.8) | 52.34 (+3.2) | 30.2 (+12.1) | 25.3 (+4.0) | 44.0 (+18.5) | 29.4 (-0.1) | 40.3 (+5.3) | | | | |
| Isaac3D | Shapes3D | ✗ | 62.8 (+26.2) | 62.6 (+28.2) | 39.8 (+37.9) | 30.5 (+10.3) | 51.9 (+35.2) | 21.8 (+28.1) | 44.9 (+27.6) | 24.3 (+6.1) | 3.6 (+16.7) | 21.2 (+8.1) | 1.5 (+5.2) |
| | | ✓ | 44.8 (+8.3) | 42.5 (+14.4) | 26.1 (+16.5) | 22.3 (+6.6) | 41.6 (+20.3) | 13.8 (+14.5) | 31.8 (+13.4) | | | | |

All FoVs in common between Source and Target are effectively classified, again except `Orientation`. As for the new FoV (`Color`), we report lower performances, but we can appreciate a significant improvement with fine-tuning if we exploit a global representation. Instead, we observe a lower improvement with the pruned representation, suggesting that the new factor is not encoded in one single dimension.

To further increase the distance between Source and Target, we consider pairs for which the semantics of the FoVs are the same, but they are different in appearance, granularity, and composition (S=Color-dSprite with T=Shapes3D, see Table 3): we can observe that even without fine-tuning, the latent representation allows the classification of the dominant FoVs of the dataset, i.e. `Floor Hue` and `Wall Hue`, also when focusing on a single dimension. Fine-tuning positively affects the average classification accuracy, especially when using the whole representation.

We finally reason on the gap between Source and Target datasets in terms of complexity. When the Source is simpler than the Target but still they have some FoVs in common, possibly with different appearances, (e.g. S=Shapes3D, T=Isaac3D, Table 9 and Table 10) we can appreciate the effectiveness of transfer and fine-tuning for all metrics. Conversely, when the Source is much more complex than the Target (e.g. when S=Isaac3D, T=Shapes3D) one could expect the richness in the Source to be directly transferrable to the simpler Target. However, we observe that the finetuning is still beneficial for all the disentanglement metrics. This can be explained by the "domain" dependence of VAE models.

Discussion. Disentanglement transfers well between synthetic datasets with the same FoVs, w.r.t. all the properties. If the Target includes new FoVs, fine-tuning is necessary for the new FoV, but also for the entire representation, as compactness and modularity are partially degraded by the new FoV. When the Source and Target become significantly different, fine-tuning is also beneficial. We can conclude that when both Source and Target are synthetic and DRL-compliant, the properties of disentangled representation are preserved before and after fine-tuning, especially when the datasets have FoVs in common even though they have different appearance.

**(2) Synthetic to Real.** We now analyse the potential of transferring a disentangled representation from an appropriately generated Synthetic Source (DRL-compliant) to a Real Target. We first consider Real

Table 4: Target dataset: Coil100 and variants (see Table 2).

| SD | TD | Pruned | Mean accuracy on FoVs(%) Object (100) | Pose (72) | Orientation (18) | Scale (9) | All | Modularity(%) Our (OS) | DCI | Compactness(%) Our (MES) | MIG |
|---|---|---|---|---|---|---|---|---|---|---|---|
| dSprites | Coil (bin) | ✗ | 13.2 (+3.9) | 1.7 (-0.1) | 48.1 (+1.7) | 38.6 (+7.4) | 25.4 (+3.2) | 37.8 (+2.6) | 11.4 (+2.7) | 25.7 (+2.1) | 5.1 (+1.8) |
| | | ✓ | 4.8 (-0.2) | 1.5 (+0.1) | 18.0 (-5.2) | 33.3 (+2.7) | 14.4 (-0.7) | | | | |
| C-dSprites | | ✗ | 23.3 (+20.9) | 1.6 (+0.1) | 34.6 (+13.2) | 38.6 (+7.6) | 24.5 (+10.4) | 33.9 (+3.0) | 10.4 (+1.8) | 27.2 (+0.0) | 4.1 (+2.2) |
| | | ✓ | 13.4 (+0.1) | 1.5 (-0.1) | 11.5 (+1.9) | 33.2 (-3.2) | 14.9 (-0.3) | | | | |
| Shapes3D | Coil | ✗ | 16.6 (+24.9) | 1.5 (+0.1) | 17.7 (+26.9) | 32.5 (+12.4) | 17.1 (+16.1) | 31.7 (-0.2) | 3.2 (+3.5) | 25.0 (-1.0) | 1.7 (+0.8) |
| | | ✓ | 6.2 (+4.6) | 1.39 (+0.0) | 9.72 (+4.5) | 23.5 (+3.6) | 10.21 (+3.2) | | | | |
| Coil(bin) | | ✗ | 20.9 (+12.6) | 1.6 (+0.0) | 36.2 (+10.0) | 30.4 (+12.8) | 22.3 (+8.8) | 36.3 (-5.2) | 10.4 (-5.3) | 26.1 (-2.4) | 5.6 (-3.5) |
| | | ✓ | 7.3 (+2.6) | 1.5 (-0.1) | 12.8 (+1.8) | 27.2 (-0.4) | 12.2 (+1.0) | | | | |

Table 5: Target dataset: RGBD-Objects and variants (see Table 2).

| SD | TD | Pruned | Mean accuracy on FoVs(%) Category (51) | Elevation (4) | Pose (263) | All | Modularity(%) Our (OS) | DCI | Compactness(%) Our (MES) | MIG |
|---|---|---|---|---|---|---|---|---|---|---|
| dSprites | RGBD (depth) | ✗ | 64.3 (+4.3) | 83.0 (+2.8) | 0.3 (+0.0) | 49.2 (+2.4) | 34.3 (+0.6) | 11.0 (+0.5) | 22.1 (-0.2) | 3.4 (+0.5) |
| | | ✓ | 34.1 (-4.3) | 67.3 (-1.5) | 0.3 (+0.0) | 33.9 (-1.9) | | | | |
| Shapes3D | | ✗ | 56.2 (+15.4) | 79.5 (+7.8) | 0.3 (+0.0) | 45.3 (+7.7) | 35.3 (-0.8) | 12.0 (-0.9) | 22.4 (-0.5) | 2.0 (+0.8) |
| | | ✓ | 25.9 (+11.7) | 65.2 (+4.8) | 0.3 (+0.0) | 30.5 (+5.5) | | | | |
| Coil(bin) | | ✗ | 17.3 (+5.5) | 58.7 (+6.7) | 0.3 (+0.0) | 25.4 (+4.1) | 35.6 (-1.5) | 11.8 (+0.3) | 23.5 (-2.0) | 4.3 (-2.4) |
| | | ✓ | 9.3 (+3.2) | 56.6 (-1.4) | 0.3 (+0.0) | 22.1 (+0.6) | | | | |
| C-dSprites | | ✗ | 86.4 (+2.5) | 63.7 (+1.3) | 0.3 (+0.0) | 50.1 (+1.3) | 35.7 (-0.7) | 7.8 (-2.3) | 24.7 (-1.5) | 1.8 (-0.4) |
| | | ✓ | 36.7 (+1.6) | 48.6 (-0.8) | 0.2 (+0.1) | 28.5 (+0.3) | | | | |
| Coil | RGBD | ✗ | 83.5 (+6.9) | 61.1 (+3.5) | 0.3 (+0.1) | 48.3 (+3.5) | 35.0 (+0.0) | 4.2 (+0.6) | 23.5 (-0.3) | 2.0 (-1.1) |
| | | ✓ | 35.7 (+0.1) | 47.6 (-0.7) | 0.3 (+0.0) | 27.8 (-0.2) | | | | |
| Coil(bin) | | ✗ | 79.3 (+9.3) | 59.2 (+5.6) | 0.3 (+0.0) | 46.3 (+5.0) | 35.3 (-0.5) | 5.1 (-0.2) | 24.2 (-1.2) | 1.3 (-0.2) |
| | | ✓ | 36.3 (+0.4) | 48.6 (-0.0) | 0.2 (+0.0) | 28.4 (+0.2) | | | | |

Targets with FoVs independence. In Table 4, we first analyse (S=Color-dSprite with T=Coil): the Target dataset shares some FoVs with the Source (such as Scale, inplane Orientation, and Object, the latter related to shape), but their variability and granularity may be very different (e.g. the shape/object can assume 3 values on Source and 100 on Target ). Other FoV are new, such as Pose (encoding 3D rotations). The accuracy we achieve is very uneven on the different FoVs, in particular Pose, since the Source does not incorporate any 3D information. We notice an improvement with fine-tuning but also a degradation in performances with the pruned representation (a sign the representation does not produce a good disentanglement on the Target). An exception to this last comment is the FoV Scale, whose accuracy does not degrade significantly with one dimension only (an indication this FoV is well represented in one dimension).

With the same Target, we assess a representation which is incorporating some level of 3D information ( S=Shapes3D, with T=Coil, Table 4). This choice does not bring any benefit since the synthetic Shapes3D includes a very simplified form of pose variation. Orientation accuracy degrades, as it is not captured by the Source (indeed, with fine tuning this performance improves). Considering the Target dataset, it clearly presents several new challenges w.r.t. Source ones, we produce a last experiment with a simplified binary version of the Target, meant to be more similar to the binary images in dSprite. In this case, some FoVs are very well represented (Orientation and Scale), while Object presents low performances due to the decrease in the image descriptive power caused by binarization.

We now consider another real Target, RGBD-Object (see Table 5). As for the former, we consider Color-dSprite as a synthetic source: here, the same considerations about the Pose we discussed before are valid, and on the other FoVs, we observe the global representation is effective (and marginally improved by fine-tuning), while the pruned representation leads to lower performances, as a sign the representation is not perfectly disentangled on the Target. Here again, we investigate the possibility of transferring to simplified versions of the dataset (in this case, we consider the Depth channel only). The FoV Elevation improves significantly as it is not directly affected by color information. Concerning Modularity (Table 4 and Table 5), fine-tuning seems to have a small influence. In particular, we observe a small degradation for RGBD-Objects (Table 5), coherent with the challenges of a real dataset with unknown/hidden factors perturbing the encoding of the FoV.

Discussion. If the Source is synthetic (and DRL-compliant), and the Target is Real, the quality of transferring seems to depend on the distance between datasets and is not even across different FoVs: FoVs that are more similar between the datasets are more easily represented, and they exhibit better compactness properties. Fine-tuning is bringing a significant benefit in Explicitness and some

(limited) benefit in Modularity and Compactness. If the Target incorporates unknown hidden factors, as we may expect to happen in the real world, Modularity and Compactness transfer worse, and the benefit of fine-tuning is limited.

**(3) Real to Real.** We conclude by discussing the possibility of transferring from a DRL-compliant real dataset to another real one. As a first task, we consider as a Source a simplified version of the Target (specifically, S=Coil-binary, with T=Coil): the source should encode the factors not related to RGB, while the finetuning should improve the disentanglement and the explicitness of the representation. However, this is not the case with Coil100, whose representations degrade the Modularity, and the finetuning only affects the entire representation.

We then consider a larger variation between Real Source and Real Target (specifically, S=Coil with T=RGBD-Object, see Table 5): we obtain similar results to those of Color-dSprites as Source Dataset (comparable Explicitness), with a reduction on the performances obtained by the pruned representation. Notice that adopting the binary Coil as a Source causes only a limited reduction in Explicitness, and this was somewhat unexpected as we have a large gap in complexity between Source and Target. Our experiments did not consider RGBD-Objects acting as a Source dataset, not being DRL-compliant.

Discussion. Using a real DRL-compliant dataset as a Source, we do not appreciate any benefit. Fine-tuning is not particularly effective. At the same time, we notice that some level of disentanglement transfer can be observed.

## 4 Limitations

A limitation of our current work is the adoption of a specific family of approaches (VAE-based). The generalization of our finding to more recent vector-based approaches (e.g. [71, 62, 48, 36]) needs further investigation. However, each family of approaches for disentanglement learning *follows specific paradigms that may require tailored designs for transfer learning*. In other words, while the general transfer methodology is still applicable, it might need proper tuning to perform optimally depending on the particular learning approach.

## 5 Conclusions

In this paper, we discussed the potential of transferring a Disentangled Representation as a strategy to address disentanglement in real data. We learned the representation from a Source Dataset in a weakly supervised manner and transfered it to a Target Dataset, where supervision on the FoVs was difficult or impossible to obtain. We identified three main scientific questions, summarised in Section 3.2, which we recall to draw conclusions on our study. Starting from question **Q2**, on the properties of disentangled representations that are preserved after transferring, we may conclude Explicitness is usually well maintained, while Modularity and Compactness are reduced as we move from synthetic to real. More precisely, we appreciate a degradation in the global metrics (such as OS and ME), while on the compactness through the analysis of the 1-dimensional pruned representations, we notice that some FoV may transfer very well.

As for **Q3**, we may observe that fine-tuning is almost always beneficial, and it never causes any harm. **Q1**, a much wider question discussing under what circumstances transfer is effective, leads us to conclude that some structural similarity between Source and Target datasets is necessary, including similar ranges/granularity of variations of related factors. A quantification of the similarity among datasets is still under investigation; *the results of our study suggest one could design synthetic data to capture/disentangle specific factors of interest*.

Future directions. Currently, we are exploring quantitative methods to assess the distance between Source and Target datasets. In the near future will target more specific applications, such as biomedical image classification or action recognition from videos, to discuss and relate the general results we are reporting in this paper to more specific and challenging domains.

## Acknowledgement

We acknowledge the financial support from PNRR MUR Project PE0000013 "Future Artificial Intelligence Research (FAIR)", funded by the European Union – NextGenerationEU, CUP J33C24000430007

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

Table 6: Summary of different metrics for disentanglement learning. L and K are the numbers of latent variables and ground truth factors, respectively.

| Metric | Intervention based | Information based | Predictor based | Classifier | #Classifiers | Measured Property |
|---|---|---|---|---|---|---|
| BetaVAE | ✓ | ✗ | ✓ | Linear/majority-vote | 1 | Modularity |
| FactorVAE | ✓ | ✗ | ✓ | Linear/majority-vote | 1 | Modularity |
| SAP | ✗ | ✗ | ✓ | Threshold value | $L \times K$ | Compactness |
| MIG | ✗ | ✓ | ✗ | None | 0 | Compactness |
| DCI | ✗ | ✗ | ✓ | LASSO / Random forest | K | Modularity Compactness Explicitness |
| Modularity | ✗ | ✓ | ✗ | None | 0 | Modularity |
| OMES(Our) | ✓ | ✓ | ✗ | None | 0 | Modularity Compactness |

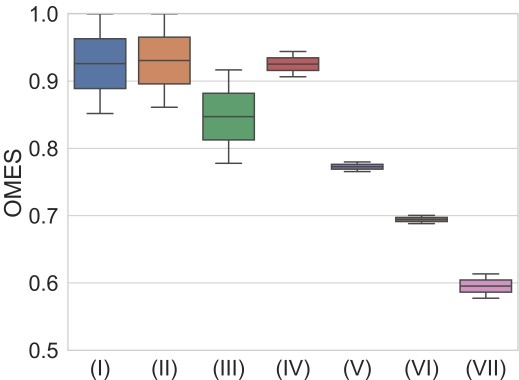

Figure 3: Boxplot of the distribution of OMES comparing 7 association matrices $S$, for different $\alpha$ values: (I), (II) and (III) are the results of simulated scenarios where only Overlap (I) and Multiple Encoding (II) or both (III) are represented, (IV), (V), (VI) are obtained with weak-supervision (respectively, on datasets Shapes3D, Color-dSprites, Noisy-dSprites; (VII) Noisy-dSprite with unsupervised model.

## A    Evaluating the quality of disentanglement

Table 6 reports the main characteristics of the well-established and most used disentanglement metrics. Note that OMES is the only one both Interventional-based and Information-based, measuring Modularity and Compactness.

## B    OMES assessment

In this section, we report the evaluation of the 1800 models trained on *Noisy-dSprites*, 1800 models trained on *SmallNORB* and 1800 models trained on *Cars3D*, all from [40]. Here we report the extensions of the results in Section 2.3

### B.1    OMES interpretation

Fig. 4 shows our metric OMES scores for different values of $\beta$ keeping the different FoV separated, for Noisy-dSprites (**Left**), SmallNORB (**Center**) and Cars3D (***Right***). $\alpha$ is fixed to 0.5.

In addition, Figure 3 shows the range of values with different $\alpha$ for a selection of S. We include 3 synthetic cases ((I), (III), (IV)) producing high scores, and (V) generated from Shapes3D, is very similar. The scores of the Noisy-dSprites models ((VI), (VII)) are lower, as it is a more challenging dataset. Color-dSprites (V) is easier to disentangle and output values in between Shapes3D and Noisy-dSprites. Note how the boxplots generated from real models produce a smaller range of values w.r.t the simulated cases; the choice of $\alpha$ does not appear critical in the real case.

## B.2 Agreement of OMES with other disentanglement metrics

Fig. 4 shows the metric scores for different values of $\beta$ keeping the different FoV with $\alpha = 0.5$, the 3 benchmark datasets (Noisy-dsprites (Left), SmallNORB (Center), Cars3D (Right)). In general the greater the $\beta$ the higher the disentanglement but the factors strictly related to reconstruction quality fail to be encoded in the representation.

Fig. 5 shows the distribution of the disentanglement metrics, extending the plot from [40] with our metric OMES computed with different values of $\alpha \in \{0.0, 0.3, 0.5, 0.8, 1.0\}$. We observe the higher the $\alpha$ the less variable the distributions of our metrics, meaning that the models are more similar in terms of *Multiple Encoding* than they are in terms of *Overlap*.

Fig. 6 shows the rank correlations of the disentanglement metrics of the models trained on *Noisy-dSprites*, extending the plot from [40] with our metric OMES computed with different values of $\alpha \in \{0.0, 0.1, 0.2, 0.3, 0.4, 0.5, 0.6, 0.7, 0.8, 0.9, 1.0\}$. We observe the higher the $\alpha$ (*Multiple Encoding*) the higher the correlations with BetaVAE Score and FactorVAE Score and negative correlations with Modularity.

Analogously, Fig. 7 shows the rank correlations of the disentanglement metrics of the models trained on *SmallNORB*, extending the plot from [40] with our metric OMES computed with different values of $\alpha \in \{0.0, 0.1, 0.2, 0.3, 0.4, 0.5, 0.6, 0.7, 0.8, 0.9, 1.0\}$.

Finally, Fig. 8 shows the rank correlations of the disentanglement metrics of the models trained on *Cars3D*, extending the plot from [40] with our metric OMES computed with different values of $\alpha \in \{0.0, 0.1, 0.2, 0.3, 0.4, 0.5, 0.6, 0.7, 0.8, 0.9, 1.0\}$.

## B.3 Agreeement with performance metrics

Fig. 9 shows Rank correlation with ELBO, reconstruction loss, and test error of FoVs classifier for the models trained on the weak-supervised setting that was shown to be more interesting for this analysis. We consider all the different Source datasets used in the transfer experiments.

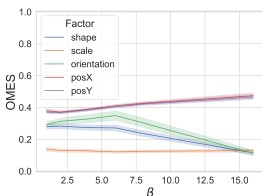 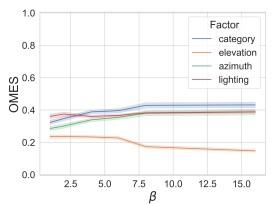 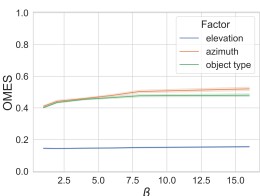

Figure 4: Scores of the proposed metric for each FoV($\alpha$ is fixed to $0.5$) of the 5400 models in [40]: 1800 models trained on Noisy-dsprites (**Left**); 1800 models trained on SmallNORB (**Center**); 1800 models trained on Cars3D (**Right**).

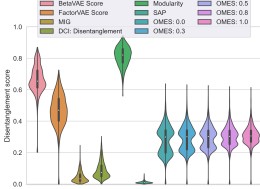 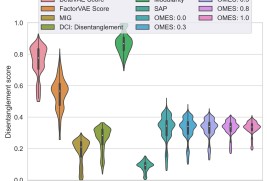 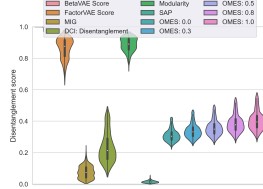

Figure 5: Distribution of different metrics of the 5400 models in [40]: 1800 models trained on Noisy-dsprites (**Left**); 1800 models trained on SmallNORB (**Center**); 1800 models trained on Cars3D (**Right**). This is an extension of the plots in [40], we added our metrics with different values of $\alpha \in \{0.0, 0.3, 0.5, 0.8, 1.0\}$.

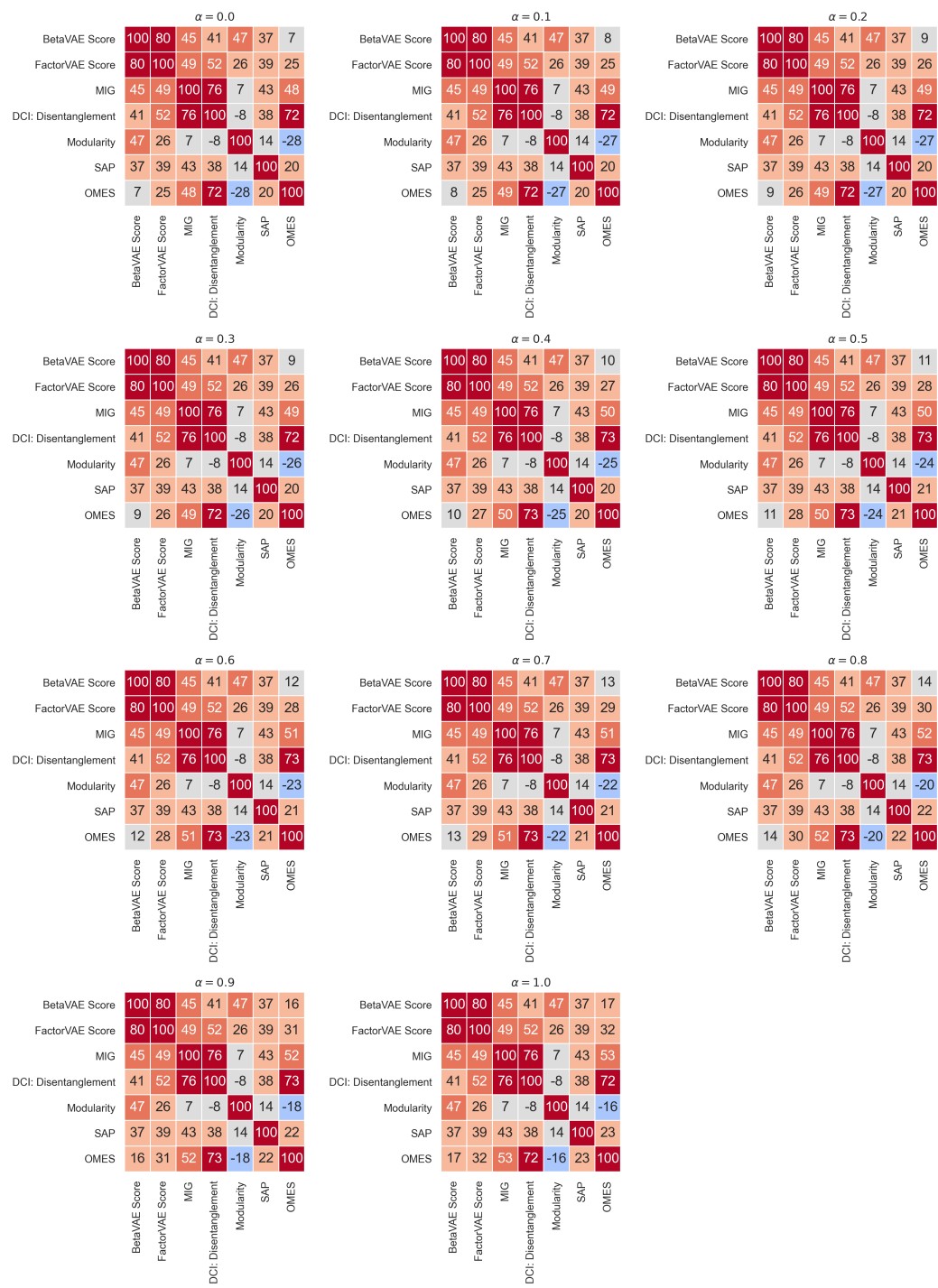

Figure 6: Rank correlation of different metrics on the same dataset (*Noisy-dSprites*) computed on the 1800 models in [40]. This is an extension of the plots in [40], we added our metrics with different values of $\alpha \in \{0.0, 0.1, 0.2, 0.3, 0.4, 0.5, 0.6, 0.7, 0.8, 0.9, 1.0\}$.

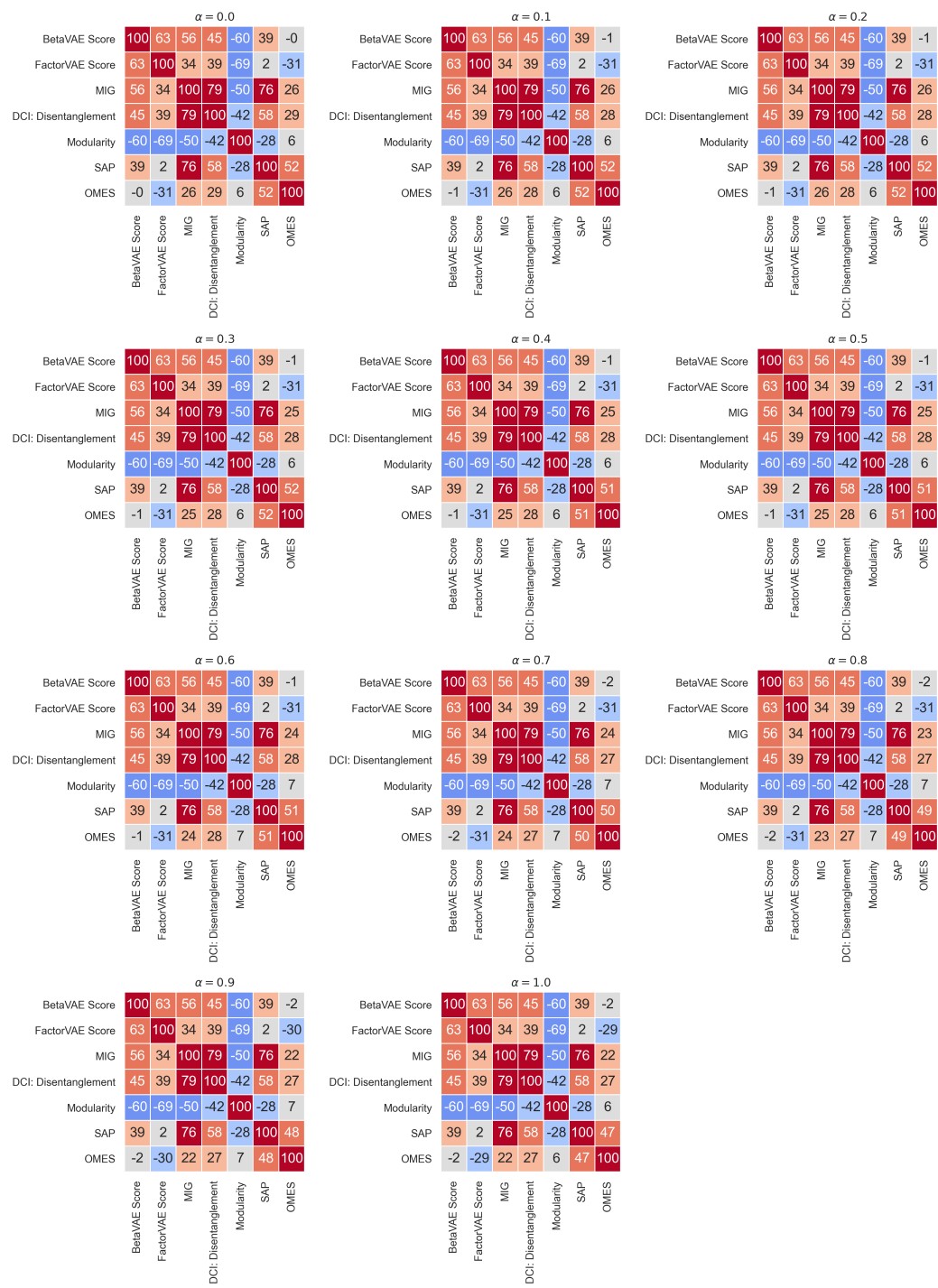

Figure 7: Rank correlation of different metrics on the same dataset (*SmallNORB*) computed on the 1800 models in [40]. This is an extension of the plots in [40], we added our metrics with different values of $\alpha \in \{0.0, 0.1, 0.2, 0.3, 0.4, 0.5, 0.6, 0.7, 0.8, 0.9, 1.0\}$.

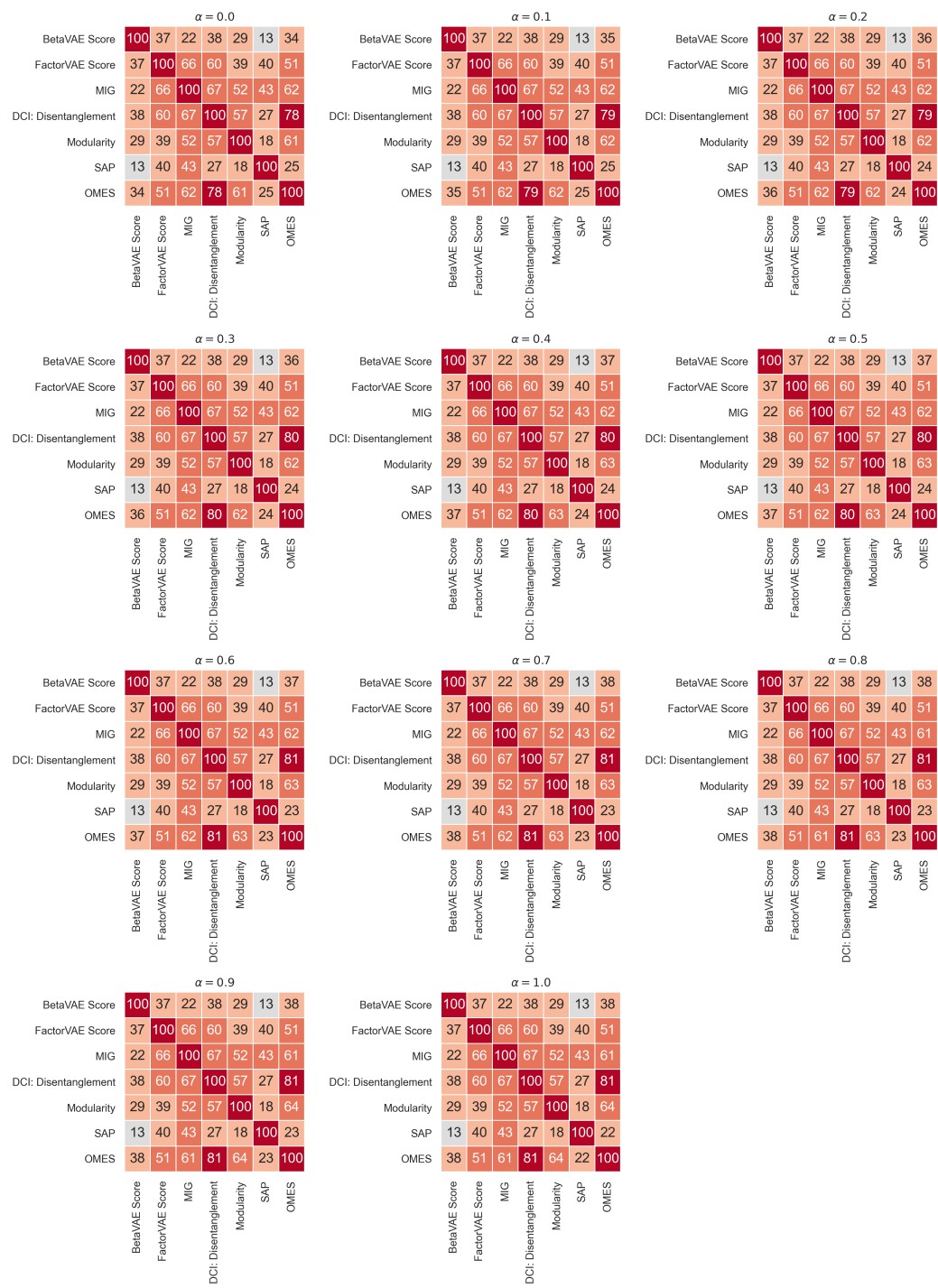

Figure 8: Rank correlation of different metrics on the same dataset (*Cars3D*) computed on the 1800 models in [40]. This is an extension of the plots in [40], we added our metrics with different values of $\alpha \in \{0.0, 0.1, 0.2, 0.3, 0.4, 0.5, 0.6, 0.7, 0.8, 0.9, 1.0\}$.

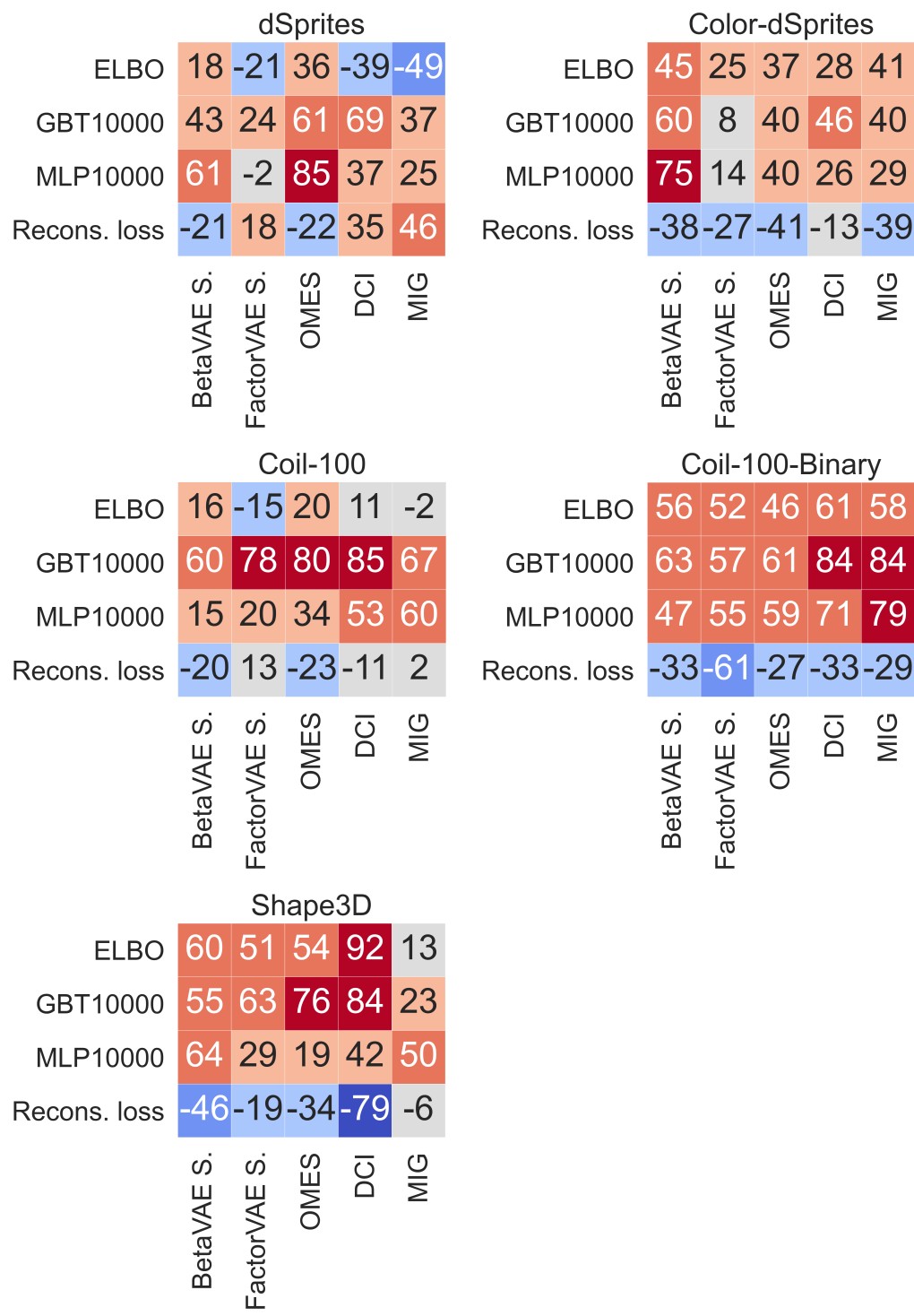

Figure 9: Rank correlations (Spearman) of ELBO, reconstruction loss, and the test accuracy of a GBT and a MLP classifier trained on 10,000 labelled data points with disentanglement metrics. In all plots OMES is computed with $\alpha = 0.5$.

Table 7: Datasets info and examples. Specifically, the *Variant* column shows the corresponding variants of the original dataset

| **Dataset FoV** | **Original** | **Variant** |
|---|---|---|

| Color-dSprites | |
|---|---|
| **FoV** | **# values** |
| Color | 7 |
| Shape | 3 |
| Scale | 6 |
| Orientation | 40 |
| PosX | 32 |
| PosY | 32 |

| Shapes3D | |
|---|---|
| **FoV** | **# values** |
| Floor Hue | 10 |
| Wall Hue | 10 |
| Object Hue | 10 |
| Scale | 8 |
| Shape | 4 |
| Orientation | 15 |

| Coil100-Augmented | |
|---|---|
| **FoV** | **# values** |
| Object | 100 |
| Pose | 72 |
| Orientation | 18 |
| Scale | 9 |

| RGBD Objects | |
|---|---|
| **FoV** | **# values** |
| Category | 51 |
| Elevation | 4 |
| Pose | 263 |

## C  Transfer experiment

Here we provide additional information about the architecture of the models for the transfer experiments. Moreover, we include the tables reporting the average performances of the GBT and MLP classifiers.

### C.1  Datasets

Table 7 reports the main information about the used dataset, e.g. FoV and number of classes, together with some examples of the original dataset, plus some samples of one variant of the given dataset, such as Color-dSprites and Noisy-Color-dSprites. The only exception is Shapes3D which does not have any variant, so different samples drawn from the same dataset are shown.

### C.2  Architecture

Table 8 shows the architecture of the model used for all the transfer experiments. We trained multiple models with 10 different random seeds for each value of $\beta$. Hence, we obtain 20 models for each *Source* dataset. We adopted the Adam optimizer [29] with default parameters, batch size=64 and 400k steps. We used linear deterministic warm-up [11, 61, 3] over the first 50k training steps. We maintained the latent dimension fixed to 10 for all the experiments.

Table 8: Encoder and Decoder architecture for the transfer experiments.

| Encoder | Decoder |
|---|---|
| **Input:** $64 \times 64 \times$ #channels | **Input:** $\mathbb{R}^{10}$ |
| $4 \times 4$ *conv*, 32 LeakyRelu(0.02), stride 2 | FC 8192 LeakyRelu(0.02) |
| $4 \times 4$ *conv*, 64 LeakyRelu(0.02), stride 2 | FC $8 \times 8 \times 128$ |
| $4 \times 4$ *conv*, 128 LeakyRelu(0.02), stride 2 | $4 \times 4$ *upconv*, 64 LeakyRelu(0.02), stride 2 |
| Flatten | $4 \times 4$ *upconv*, 32 LeakyRelu(0.02), stride 2 |
| $2 \times$ FC 10 | $4 \times 4$ *upconv*, #channels Sigmoid, stride 2 |

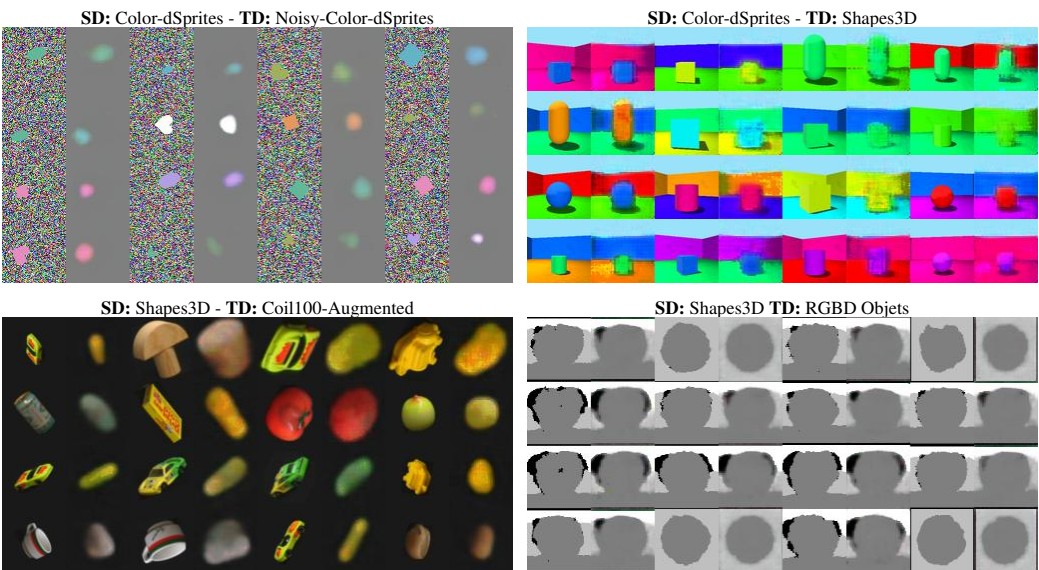

Figure 10: Some reconstructions generated from the fine-tuned models of different Source (SD) Target (TD) couples.

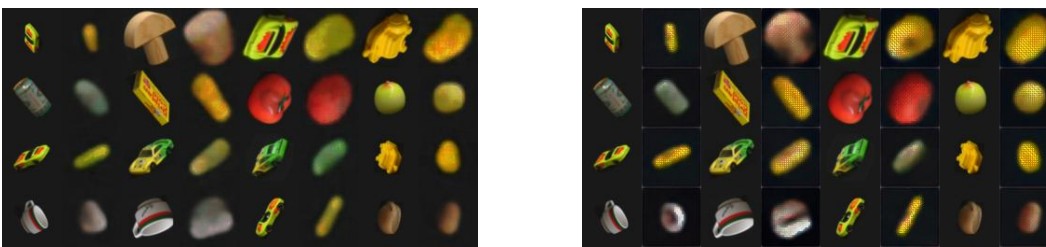

Figure 11: (**Left**) Reconstruction of samples of Coil100 of a fine-tuned model trained originally on Color-dSprites, same as in Fig. 10. (**Right**) Reconstruction of samples of Coil100 of a model trained from scratch on it.

### C.3 Transfer reconstruction

Some reconstructions, generated by the VAE models **after** they are fine-tuned, are depicted in Fig. 10. The quality of the reconstruction is good even if the encoding is obtained by training the model first on a completely different source dataset and then fine-tuning the model for a few iterations.

Fig. 11 (Right) shows the reconstruction of the same samples of Coil100-Augmented from the model trained from scratch on it. Comparing the latter with the reconstructions of Fig. 11(Left) it can be observed that the quality is comparable (with some exceptions), and so with the fine-tuned models, we are not losing much information from the data.

Table 9: Transfer from Shapes3D (Source) to Isaac3D (Target). Average classification *accuracy* over the 20 models of the GBT classifier, before and after fine-tuning (see Table 2).

| Pruned | Object shape (3) | Object scale (4) | Camera height (4) | X-movement (8) | Y-movement (5) | Light intensity (4) | Light y-direction (6) | Object color (4) | Wall color (4) | All |
|--------|---------|---------|---------|---------|---------|---------|---------|---------|---------|-----|
| ✗ | 34.9 (+5.1) | 54.0 (+34.6) | 39.2 (+17.4) | 33.9 (+29.6) | 23.0 (+4.3) | 83.6 (+14.0) | 85.4 (+12.7) | 29.8 (+13.9) | 78.1 (+18.9) | 51.3 (+16.7) |
| ✓ | 33.8 (+3.6) | 40.1 (+24.1) | 33.5 (+11.1) | 24.7 (+15.3) | 21.6 (+2.7) | 69.4 (+17.3) | 67.5 (+14.4) | 27.0 (+7.3) | 61.5 (+16.6) | 42.1 (+12.5) |

Table 10: Transfer from Shapes3D (Source) to Isaac3D (Target). The *Compactness* and *Modularity* scores of the same models of Table 9.

| Pruned | Modularity(%) | | Compactness(%) | |
|--------|-------------|-----|-------------|-----|
| | Our (OS) | DCI | Our (MES) | MIG |
| ✗ | 25.1 | 6.3 | 21.2 | 2.2 |
| ✓ | (+9.7) | (+16.1) | (+10.2) | (+5.9) |

## C.4 Transfer protocol

Each GBT and MLP classifiers are trained on the latent representation $r$ extracted from the Encoder of the Ada-GVAE models so that the train split comprises 10000 samples and the test split is of 5000 samples.

With *unsupervised* fine-tuning on the Target dataset, it means that the model is trained as a simple VAE [30].

## C.5 GBT & MLP performance distribution

In this section, we report the performance distribution of the GBT classifiers on the target FoV (see Fig. 12), the figures on the left depict the performances on the representation before fine-tuning and on the right are depicted the results after the fine-tuning of the representation. Fig. 13 shows the performance distribution of the MLP classifiers on the target FoV, the figures on the left depict the performances on the representation before fine-tuning and on the right depict the results after the fine-tuning of the representation.

## C.6 Transfer from Shapes3D to Isaac3D

In this section, we report the results of the transfer from Shapes3D to Isaac3D, see Table 9 for the Explicitness and Table 10 for Modularity and Compactness.

We observe that the transfer is overall effective according to all the disentanglement scores even though the Target dataset is much more complex than the Source. If we compare the results of the transfer from Shapes3D to other real datasets such as Coil100 (Table 4) and RGBD-Objects (Table 5), we notice that in the case of Isaac3D the boost of finetuning is more noticeable.

This suggests that, if source and target have common FoVs with similar appearance we can obtain reasonable performances on a real Target dataset even if the source synthetic dataset is much simpler.

## C.7 MLPs performances

In this section, we report the performances of the MLP classifiers on the FoVs, for the sake of comparison we report the scores of the disentanglement metrics on the representation input.

We associate the tables regarding the same Target dataset: Table 11 corresponds to the same representations of Table 2 in the main document; Table 12 to Table 3; Table 13 to Table 4 and Table 14 to Table 5.

Overall we can observe higher performances obtained with MLP, especially on the classifiers trained on the entire representation. This happens because MLP can easily disentangle an entangled representation by observing different dimensions at a time while GBTs partition the input space into regions aligned with the axes, making it harder to observe multiple dimensions at a time and so the FoVs.

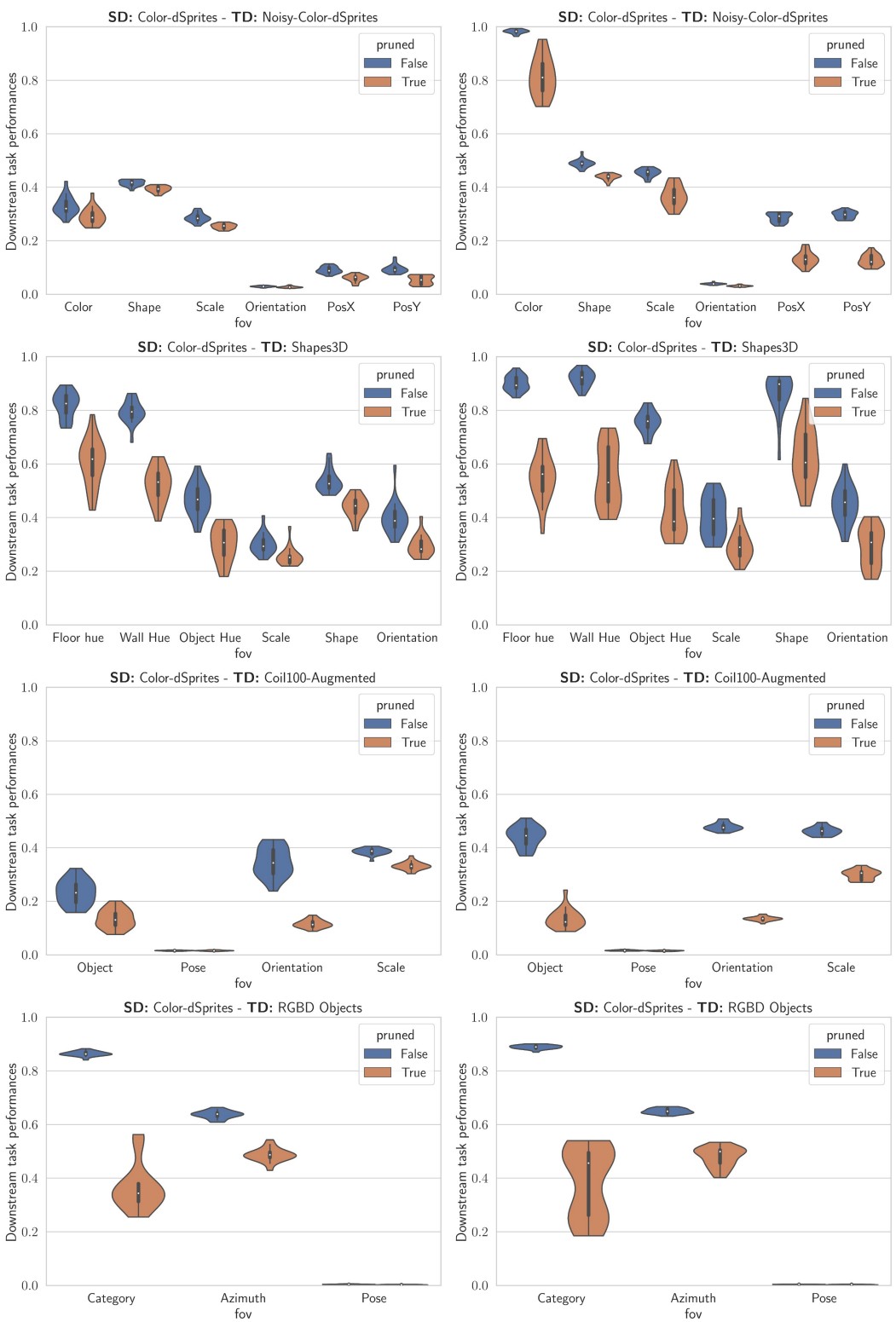

Figure 12: Some examples of the performance distribution of the GBT classifiers before (**Left**) and after (**Right**) fine-tuning of different Source (SD) Target (TD) couples.

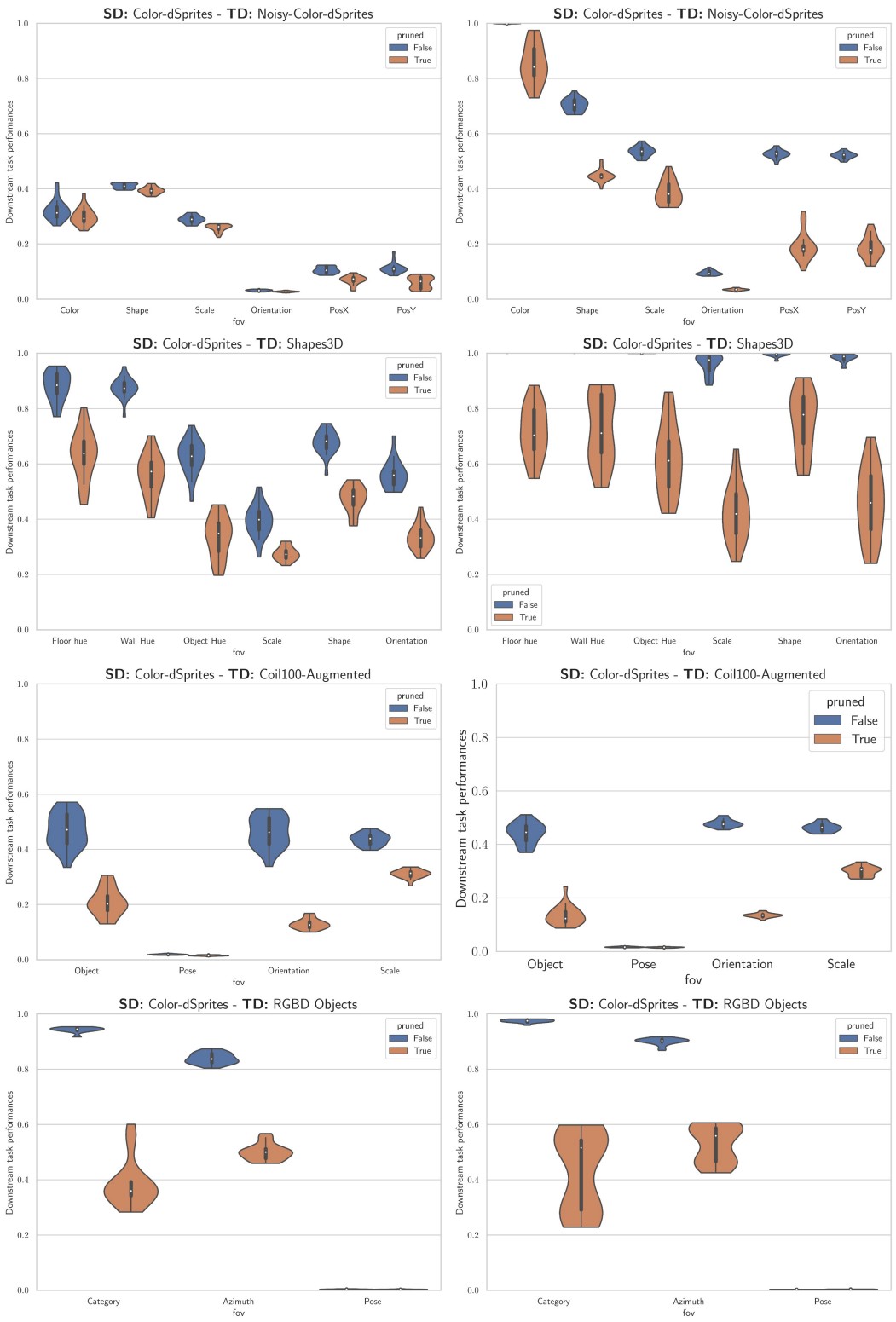

Figure 13: Some examples of the performance distribution of the MLP classifiers before (**Left**) and after (**Right**) fine-tuning of different Source (SD) Target (TD) couples.

Table 11: Target dataset: dSprites and variant. The averaged performances of the MLP classifiers and disentanglement metrics, together with the average improvement with the finetuning (in brackets).

| SD | TD | Pruned | Color (7) | Shape (3) | Scale (6) | Orientation (40) | PosX (32) | PosY (32) | All | Our (OS) | DCI | Our (MES) | MIG |
|---|---|---|---|---|---|---|---|---|---|---|---|---|---|
| | | | | | | Mean Accuracy on FoVs(%) | | | | Modularity(%) | | Compactness(%) | |
| dSprites | N-dSprites | ✗ | | 64.1 (+33.2) | 44.3 (+45.9) | 8.7 (+33.7) | 20.1 (+54.0) | 19.4 (+54.0) | 31.3 (+44.2) | 31.8 (+6.3) | 22.0 (+4.0) | 32.1 (+3.9) | 13.9 (+6.9) |
| | | ✓ | | 50.9 (+14.3) | 42.9 (+22.3) | 4.1 (+8.8) | 15.4 (+27.7) | 15.9 (+26.7) | 25.8 (+20.0) | | | | |
| | C-dSprites | ✗ | 39.3 (+45.0) | 99.4 (+0.2) | 96.3 (+2.1) | 67.6 (+5.9) | 85.2 (-1.9) | 83.6 (-2.5) | 78.6 (+8.1) | 61.0 (+1.6) | 42.9 (+10.4) | 72.3 (+3.5) | 34.0 (+0.3) |
| | | ✓ | 19.3 (+13.1) | 74.3 (+5.6) | 78.2 (+6.4) | 21.1 (+3.1) | 73.8 (-1.8) | 73.4 (-1.6) | 56.7 (+4.1) | | | | |
| C-dSprites | N-C-dSprites | ✗ | 32.2 (+67.7) | 41.0 (+29.4) | 28.9 (+24.6) | 3.1 (+6.4) | 10.5 (+42.1) | 11.1 (+41.0) | 21.1 (+35.2) | 28.9 (+0.3) | 5.4 (+2.2) | 29.0 (-1.9) | 1.8 (+1.0) |
| | | ✓ | 29.9 (+55.4) | 39.4 (+5.3) | 25.8 (+12.9) | 2.7 (+0.7) | 6.8 (+12.4) | 6.0 (+12.6) | 18.4 (+16.6) | | | | |

Table 12: Target dataset: Shapes3D. The averaged performances of the MLP classifiers and disentanglement metrics, together with the average improvement with the finetuning (in brackets).

| SD | TD | Pruned | Floor Hue (10) | Wall Hue (10) | Object Hue (10) | Scale (8) | Shape (4) | Orientation (15) | All | Our (OS) | DCI | Our (MES) | MIG |
|---|---|---|---|---|---|---|---|---|---|---|---|---|---|
| | | | | | | Mean Accuracy on FoVs(%) | | | | Modularity(%) | | Compactness(%) | |
| dSprites | | ✗ | 84.7 (+15.3) | 86.8 (+13.2) | 57.3 (+40.3) | 33.4 (+55.8) | 67.5 (+27.9) | 51.1 (+44.9) | 63.4 (+32.9) | 26.9 (+0.6) | 9.6 (+9.0) | 23.9 (+1.1) | 5.3 (-0.4) |
| | | ✓ | 64.1 (+14.6) | 61.3 (+18.4) | 31.9 (+33.1) | 23.1 (+16.7) | 45.2 (+23.4) | 27.0 (+13.4) | 42.1 (+19.9) | | | | |
| C-dSprites | Shapes3D | ✗ | 88.1 (+11.9) | 87.4 (+12.6) | 62.5 (+37.5) | 39.5 (+56.6) | 67.7 (+31.6) | 56.0 (+42.3) | 66.9 (+32.1) | 30.6 (+0.8) | 13.7 (+7.4) | 28.2 (+2.1) | 8.4 (-1.4) |
| | | ✓ | 63.2 (+8.3) | 55.9 (+16.6) | 33.7 (+27.8) | 27.5 (+14.6) | 47.2 (+27.7) | 33.6 (+12.0) | 43.5 (+17.8) | | | | |

This aligns with what is observed in [41, 11]. Given this, it is preferable to observe the performances of the GBT because they can give a clearer about the disentanglement properties of the representation.

## C.8    GBT & MLP performances standard deviation

In this section, we report the standard deviation of performances of the GBT and MLP classifiers on the FoVs, for the sake of comparison we also report the standard deviation of scores of the disentanglement metrics on the representation input.

**MLP classifiers**: Table 19 corresponds to the same representations of Table 11 in the Appendix; Table 20 to Table 12; Table 21 to Table 13 and Table 22 to Table 14.

**GBT classifiers**: Table 15 corresponds to the same representations of Table 2 in the Appendix; Table 16 to Table 3; Table 17 to Table 4 and Table 18 to Table 5.
Note that the metrics scores in the MLPs and GBTs tables are the same because we refer to the same representations. In general, we can observe the models without fine-tuning have a small standard deviation, while fine-tuned models perform with higher variation.

Table 13: Target dataset: Coil100-augmented and variants. The averaged performances of the MLP classifiers and disentanglement metrics, together with the average improvement with the finetuning (in brackets).

| SD | TD | Pruned | Object (100) | Pose (72) | Orientation (18) | Scale (9) | All | Our (OS) | DCI | Our (MES) | MIG |
|---|---|---|---|---|---|---|---|---|---|---|---|
| | | | | | Mean Accuracy on FoVs(%) | | | Modularity(%) | | Compactness(%) | |
| dSprites | Coil (bin) | ✗ | 37.2 (+25.7) | 2.6 (+0.5) | 55.6 (+13.3) | 44.6 (+21.6) | 35.0 (+15.3) | 37.8 (+2.6) | 11.4 (+2.7) | 25.7 (+2.1) | 5.1 (+1.8) |
| | | ✓ | 9.2 (-0.6) | 1.7 (-0.1) | 19.5 (-5.0) | 34.5 (+1.3) | 16.2 (-1.1) | | | | |
| C-dSprites | | ✗ | 46.9 (+41.1) | 1.9 (+0.8) | 46.3 (+22.9) | 43.6 (+23.4) | 34.7 (+22.1) | 33.9 (+3.0) | 10.4 (+1.8) | 27.2 (+0.0) | 4.1 (+2.2) |
| | | ✓ | 20.8 (-0.6) | 1.5 (+0.2) | 12.9 (+2.3) | 31.2 (-0.2) | 16.6 (+0.4) | | | | |
| Shapes3D | Coil | ✗ | 32.0 (+58.3) | 1.7 (+1.0) | 27.3 (+41.6) | 36.0 (+30.3) | 24.2 (+32.8) | 31.7 (-0.2) | 3.2 (+3.5) | 25.0 (-1.0) | 1.7 (+0.8) |
| | | ✓ | 10.1 (+8.6) | 1.5 (+0.1) | 11.1 (+6.2) | 24.1 (+3.7) | 11.7 (+4.7) | | | | |
| Coil(bin) | | ✗ | 51.3 (+34.5) | 2.5 (+0.2) | 51.1 (+18.8) | 35.1 (+31.6) | 35.0 (+21.3) | 36.3 (-5.2) | 10.4 (-5.3) | 26.1 (-2.4) | 5.6 (-3.5) |
| | | ✓ | 12.5 (+5.6) | 1.7 (-0.1) | 14.7 (+3.0) | 28.1 (-0.2) | 14.3 (+2.1) | | | | |

Table 14: Target dataset: RGBD-Objects and variants. The averaged performances of the MLP classifiers and disentanglement metrics, together with the average improvement with the finetuning (in brackets).

| SD | TD | Pruned | Category (51) | Elevation (4) | Pose (263) | All | Our (OS) | DCI | Our (MES) | MIG |
|---|---|---|---|---|---|---|---|---|---|---|
| | | | Mean Accuracy on FoVs(%) | | | | Modularity(%) | | Compactness(%) | |
| dSprites | RGBD (depth) | ✗ | 73.9 (+9.3) | 90.3 (+4.5) | 0.3 (-0.1) | 54.8 (+4.6) | 34.3 (+0.6) | 11.0 (+0.5) | 22.1 (-0.2) | 3.4 (+0.5) |
| | | ✓ | 36.4 (-2.9) | 68.6 (-0.5) | 0.3 (+0.0) | 35.1 (-1.1) | | | | |
| Shapes3D | | ✗ | 70.5 (+15.0) | 88.4 (+7.5) | 0.3 (-0.1) | 53.0 (+7.5) | 35.3 (-0.8) | 12.0 (-0.9) | 22.4 (-0.5) | 2.0 (+0.8) |
| | | ✓ | 28.4 (+13.5) | 66.7 (+6.4) | 0.3 (-0.0) | 31.8 (+6.6) | | | | |
| Coil(bin) | | ✗ | 20.8 (+6.2) | 62.8 (+6.7) | 0.3 (+0.0) | 28.0 (+4.3) | 35.6 (-1.5) | 11.8 (+0.3) | 23.5 (-2.0) | 4.3 (-2.4) |
| | | ✓ | 10.2 (+3.8) | 57.7 (-2.3) | 0.3 (+0.0) | 22.7 (+0.5) | | | | |
| C-dSprites | RGBD | ✗ | 94.3 (+3.2) | 83.9 (+6.2) | 0.3 (-0.1) | 59.5 (+3.1) | 35.7 (-0.7) | 7.8 (-2.3) | 24.7 (-1.5) | 1.8 (-0.4) |
| | | ✓ | 38.6 (+4.1) | 50.0 (+2.9) | 0.3 (+0.0) | 29.6 (+2.3) | | | | |
| Coil | RGBD | ✗ | 95.6 (+2.7) | 89.7 (+2.1) | 0.2 (+0.0) | 61.8 (+1.6) | 35.0 (+0.0) | 4.2 (+0.6) | 23.5 (-0.3) | 2.0 (-1.1) |
| | | ✓ | 41.2 (-1.0) | 52.7 (-1.6) | 0.3 (+0.0) | 31.4 (-0.8) | | | | |
| Coil(bin) | | ✗ | 93.7 (+4.1) | 86.6 (+3.8) | 0.2 (+0.0) | 60.2 (+2.6) | 35.3 (-0.5) | 5.1 (-0.2) | 24.2 (-1.2) | 1.3 (-0.2) |
| | | ✓ | 40.6 (+1.2) | 52.5 (+1.1) | 0.3 (+0.1) | 31.1 (+0.8) | | | | |

Table 15: Target dataset: dSprites and variant. The standard deviation of the performances of the GBT classifiers and disentanglement metrics, together with the standard deviation of the performances with the finetuning (in brackets).

| SD | TD | Pruned | Color (7) | Shape (3) | Scale (6) | Orientation (40) | PosX (32) | PosY (32) | Our (OS) | DCI | Our (MES) | MIG |
|---|---|---|---|---|---|---|---|---|---|---|---|---|
| | | | Standard deviation Accuracy on FoVs(%) | | | | | | Modularity(%) | | Compactness(%) | |
| dSprites | N-dSprites | ✗ | | 0.02 (8.29) | 0.02 (6.00) | 0.01 (5.55) | 0.01 (6.65) | 0.02 (7.73) | 0.01 (3.43) | 0.01 (7.23) | 0.01 (2.99) | 0.01 (6.29) |
| | | ✓ | | 0.06 (8.65) | 0.02 (6.98) | 0.01 (4.67) | 0.01 (12.33) | 0.01 (13.49) | | | | |
| | C-dSprites | ✗ | 0.05 (9.44) | 0.02 (1.31) | 0.04 (1.89) | 0.06 (3.02) | 0.01 (1.52) | 0.01 (1.68) | 0.01 (0.67) | 0.01 (1.93) | 0.02 (1.02) | 0.01 (2.99) |
| | | ✓ | 0.03 (19.28) | 0.05 (3.54) | 0.05 (2.85) | 0.04 (5.67) | 0.01 (0.93) | 0.01 (1.40) | | | | |
| C-dSprites | N-C-dSprites | ✗ | 0.03 (0.78) | 0.01 (1.53) | 0.02 (1.53) | 0.00 (0.33) | 0.01 (1.72) | 0.02 (1.41) | 0.01 (2.09) | 0.01 (2.99) | 0.01 (2.44) | 0.01 (2.00) |
| | | ✓ | 0.03 (7.33) | 0.01 (1.18) | 0.01 (3.99) | 0.00 (0.31) | 0.01 (2.60) | 0.02 (2.15) | | | | |

# D   Experiments Compute Resources

All the experiments have been executed with an NVIDIA Quadro RTX 6000. On average, the training and evaluation of a single Source model take 5 hours. Each fine-tuning and final evaluation takes 1.5 hours. Overall, the whole bunch of transfer experiments and our metric assessment take approximately 1100 hours.

Table 16: Target dataset: Shapes3D. The standard deviation of the performances of the GBT classifiers and disentanglement metrics, together with the standard deviation of the performances with the finetuning (in brackets).

| SD | TD | Pruned | Floor Hue (10) | Wall Hue (10) | Object Hue (10) | Scale (8) | Shape (4) | Orientation (15) | Our (OS) | DCI | Our (MES) | MIG |
|---|---|---|---|---|---|---|---|---|---|---|---|---|
| | | | Standard deviation Accuracy on FoVs(%) | | | | | | Modularity(%) | | Compactness(%) | |
| dSprites | Shapes3D | ✗ | 0.05 (2.77) | 0.05 (3.08) | 0.05 (9.73) | 0.02 (6.58) | 0.05 (13.89)) | 0.06 (6.59) | 0.02 (5.60) | 0.01 (4.93) | 0.02 (4.52) | 0.01 (2.13) |
| | | ✓ | 0.06 (10.43) | 0.06 (11.44) | 0.04 (6.40) | 0.02 (5.57) | 0.04 (11.17) | 0.04 (6.71) | | | | |
| C-dSprites | | ✗ | 0.09 (8.43) | 0.07 (11.10) | 0.06 (9.17) | 0.03 (5.53) | 0.04 (10.63) | 0.04 (7.35) | 0.02 (3.82) | 0.02 (4.57) | 0.02 (3.82) | 0.01 (2.72) |
| | | ✓ | 0.09 (8.43) | 0.07 (11.10) | 0.06 (9.17) | 0.03 (5.53) | 0.04 (10.63) | 0.04 (7.35) | | | | |

Table 17: Target dataset: Coil100-augmented and variants. The standard deviation of the performances of the GBT classifiers and disentanglement metrics, together with the standard deviation of the performances with the finetuning (in brackets).

| SD | TD | Pruned | Standard deviation Accuracy on FoVs(%) | | | | Modularity(%) | | Compactness(%) | |
| | | | Object (100) | Pose (72) | Orientation (18) | Scale (9) | Our (OS) | DCI | Our (MES) | MIG |
|---|---|---|---|---|---|---|---|---|---|---|
| dSprites | Coil (bin) | ✗ | 0.01 (0.72) | 0.00 (0.20) | 0.01 (0.69) | 0.01 (0.58) | 0.00 (0.67) | 0.00 (0.51) | 0.00 (0.48) | 0.00 (0.51) |
| | | ✓ | 0.00 (0.51) | 0.00 (0.13) | 0.02 (0.54) | 0.01 (1.15) | | | | |
| C-dSprites | | ✗ | 0.05 (3.85) | 0.00 (0.17) | 0.05 (1.44) | 0.01 (1.56) | 0.00 (0.48) | 0.00 (0.51) | 0.00 (0.48) | 0.00 (0.51) |
| | | ✓ | 0.03 (3.56) | 0.00 (0.15) | 0.01 (0.81) | 0.01 (1.87) | | | | |
| Shapes3D | Coil | ✗ | 0.02 (3.12) | 0.00 (0.18) | 0.01 (1.80) | 0.01 (1.30) | 0.01 (1.35) | 0.01 (1.41) | 0.01 (1.07) | 0.01 (1.19) |
| | | ✓ | 0.01 (3.94) | 0.00 (0.12) | 0.01 (2.02) | 0.01 (2.98) | | | | |
| Coil(bin) | | ✗ | 0.03 (2.24) | 0.00 (0.14) | 0.02 (1.48) | 0.02 (1.52) | 0.01 (1.62) | 0.02 (1.52) | 0.01 (1.17) | 0.01 (1.03) |
| | | ✓ | 0.01 (3.94) | 0.00 (0.18) | 0.02 (2.57) | 0.02 (3.51) | | | | |

Table 18: Target dataset: RGBD-Objects and variants. The standard deviation of the performances of the GBT classifiers and disentanglement metrics, together with the standard deviation of the performances with the finetuning (in brackets).

| SD | TD | Pruned | Standard deviation Accuracy on FoVs(%) | | | Modularity(%) | | Compactness(%) | |
| | | | Category (51) | Elevation (4) | Pose (263) | Our (OS) | DCI | Our (MES) | MIG |
|---|---|---|---|---|---|---|---|---|---|
| dSprites | RGBD (depth) | ✗ | 0.01 (1.79) | 0.02 (0.66) | 0.00 (0.11) | 0.01 (0.47) | 0.02 (4.03) | 0.01 (0.54) | 0.01 (2.47) |
| | | ✓ | 0.05 (5.58) | 0.05 (4.65) | 0.00 (0.10) | | | | |
| Shapes3D | | ✗ | 0.02 (0.96) | 0.02 (0.59) | 0.00 (0.11) | 0.01 (0.54) | 0.02 (4.03) | 0.01 (0.54) | 0.01 (2.47) |
| | | ✓ | 0.05 (6.07) | 0.04 (5.60) | 0.00 (0.07) | | | | |
| Coil(bin) | | ✗ | 0.27 (32.60) | 0.13 (15.39) | 0.00 (0.10) | 0.01 (0.53) | 0.05 (5.73) | 0.01 (0.57) | 0.02 (1.31) |
| | | ✓ | 0.14 (18.09) | 0.11 (11.46) | 0.00 (0.08) | | | | |
| C-dSprites | RGBD | ✗ | 0.01 (0.75) | 0.01 (0.99) | 0.00 (0.11) | 0.00 (0.39) | 0.02 (2.39) | 0.01 (0.77) | 0.01 (0.85) |
| | | ✓ | 0.09 (12.79) | 0.02 (3.64) | 0.00 (0.10) | | | | |
| Coil | | ✗ | 0.01 (0.77) | 0.01 (1.02) | 0.00 (0.11) | 0.00 (0.36) | 0.01 (2.92) | 0.01 (0.67) | 0.01 (0.62) |
| | | ✓ | 0.04 (11.12) | 0.02 (2.88) | 0.00 (0.11) | | | | |
| Coil(bin) | | ✗ | 0.02 (0.70) | 0.01 (0.83) | 0.00 (0.11) | 0.00 (0.42) | 0.03 (2.43) | 0.01 (0.90) | 0.00 (0.64) |
| | | ✓ | 0.10 (12.38) | 0.03 (3.76) | 0.00 (0.08) | | | | |

# E  Future directions

We briefly summarise the future direction of our work.

On the data set side, we only considered the synthetic datasets applicable to a wider number of tasks, other synthetic datasets exist but either they are limited in the number of FoV or very specific to a task. We limited our analysis to two real datasets, plus their simplified variants, to tackle a broad but limited number of challenges. We will extend our analysis to more complex real datasets with an increasing number of known and unknown factors, but this requires we also design more complex synthetic datasets able to tackle these complications. For the methods, we will explore the effect of the dimensions of the latent space and different kinds of supervision for training the Source model, as well as including (partial) supervision on the fine-tuning. On the applications side, we will analyse

Table 19: Target dataset: dSprites and variant. The standard deviation of the performances of the MLP classifiers and disentanglement metrics, together with the standard deviation of the performances with the finetuning (in brackets).

| SD | TD | Pruned | STD Accuracy on FoVs(%) | | | | | | Modularity(%) | | Compactness(%) | |
| | | | Color (7) | Shape (3) | Scale (6) | Orientation (40) | PosX (32) | PosY (32) | Our (OS) | DCI | Our (MES) | MIG |
|---|---|---|---|---|---|---|---|---|---|---|---|---|
| dSprites | N-dSprites | ✗ | | 0.02 (1.36) | 0.03 (3.41) | 0.01 (7.36) | 0.01 (4.13) | 0.02 (5.05) | 0.01 (3.43) | 0.01 (7.23) | 0.01 (2.99) | 0.01 (6.29) |
| | | ✓ | | 0.06 (10.38) | 0.03 (12.67) | 0.01 (7.66) | 0.02 (13.12) | 0.01 (14.77) | | | | |
| | C-dSprites | ✗ | 0.08 (5.14) | 0.00 (0.27) | 0.01 (0.60) | 0.06 (3.49) | 0.05 (5.27) | 0.03 (4.01) | 0.01 (0.67) | 0.01 (1.93) | 0.02 (1.02) | 0.01 (2.99) |
| | | ✓ | 0.02 (21.72) | 0.06 (3.62) | 0.06 (3.71) | 0.04 (7.10) | 0.02 (1.54) | 0.02 (1.54) | | | | |
| C-dSprites | N-C-dSprites | ✗ | 0.04 (0.08) | 0.01 (2.34) | 0.01 (1.84) | 0.00 (0.82) | 0.01 (1.53) | 0.02 (1.18) | 0.01 (2.09) | 0.01 (2.99) | 0.01 (2.44) | 0.01 (2.00) |
| | | ✓ | 0.03 (7.10) | 0.01 (2.13) | 0.01 (4.27) | 0.00 (0.40) | 0.02 (5.12) | 0.02 (4.01) | | | | |

Table 20: Target dataset: Shapes3D. The standard deviation of the performances of the MLP classifiers and disentanglement metrics, together with the standard deviation of the performances with the finetuning (in brackets).

| SD | TD | Pruned | STD Accuracy on FoVs(%) | | | | | | Modularity(%) | | Compactness(%) | |
|---|---|---|---|---|---|---|---|---|---|---|---|---|
| | | | Floor Hue (10) | Wall Hue (10) | Object Hue (10) | Scale (8) | Shape (4) | Orientation (15) | Our (OS) | DCI | Our (MES) | MIG |
| dSprites | Shapes3D | ✗ | 0.06 (0.03) | 0.04 (0.02) | 0.07 (5.80) | 0.02 (13.47) | 0.04 (6.77) | 0.06 (4.87) | 0.02 (5.60) | 0.01 (4.93) | 0.02 (4.52) | 0.01 (2.13) |
| | | ✓ | 0.05 (7.33) | 0.07 (9.73) | 0.05 (12.26) | 0.03 (12.59) | 0.04 (13.96) | 0.05 (14.91) | | | | |
| C-dSprites | | ✗ | 0.05 (0.00) | 0.04 (0.00) | 0.06 (0.02) | 0.06 (3.56) | 0.04 (0.78) | 0.05 (1.46) | 0.02 (3.82) | 0.02 (4.57) | 0.02 (3.82) | 0.01 (2.72) |
| | | ✓ | 0.09 (9.44) | 0.08 (12.63) | 0.08 (12.60) | 0.02 (10.19) | 0.05 (11.03) | 0.04 (12.73) | | | | |

Table 21: Target dataset: Coil100-augmented and variants. The standard deviation of the performances of the MLP classifiers and disentanglement metrics, together with the standard deviation of the performances with the finetuning (in brackets).

| SD | TD | Pruned | STD Accuracy on FoVs(%) | | | | Modularity(%) | | Compactness(%) | |
|---|---|---|---|---|---|---|---|---|---|---|
| | | | Object (100) | Pose (72) | Orientation (18) | Scale (9) | Our (OS) | DCI | Our (MES) | MIG |
| dSprites | Coil (bin) | ✗ | 0.02 (0.90) | 0.00 (0.25) | 0.01 (0.67) | 0.02 (1.21) | 0.00 (0.67) | 0.00 (0.51) | 0.00 (0.48) | 0.00 (0.51) |
| | | ✓ | 0.01 (0.69) | 0.00 (0.17) | 0.01 (0.70) | 0.01 (0.94) | | | | |
| C-dSprites | | ✗ | 0.02 (0.90) | 0.00 (0.25) | 0.01 (0.67) | 0.02 (1.21) | 0.00 (0.48) | 0.00 (0.51) | 0.00 (0.48) | 0.00 (0.51) |
| | | ✓ | 0.01 (0.69) | 0.00 (0.17) | 0.01 (0.70) | 0.01 (0.94) | | | | |
| Shapes3D | Coil | ✗ | 0.02 (0.91) | 0.00 (0.26) | 0.01 (0.77) | 0.02 (1.16) | 0.01 (1.35) | 0.01 (1.41) | 0.01 (1.07) | 0.01 (1.19) |
| | | ✓ | 0.01 (6.04) | 0.00 (0.18) | 0.01 (3.44) | 0.01 (3.44) | | | | |
| Coil(bin) | | ✗ | 0.04 (1.65) | 0.00 (0.19) | 0.03 (0.85) | 0.02 (2.02) | 0.01 (1.62) | 0.02 (1.52) | 26.1 (-2.4) | 0.01 (1.03) |
| | | ✓ | 0.03 (6.86) | 0.00 (0.21) | 0.02 (3.56) | 0.02 (4.13) | | | | |

the effect of transferring from different priors (e.g. video sequences, expert knowledge in biological data) and investigate if transferring a disentangled representation will help to increase the level of interpretability of the target representation.

Table 22: Target dataset: RGBD-Objects and variants. The standard deviation of the performances of the MLP classifiers and disentanglement metrics, together with the standard deviation of the performances with the finetuning (in brackets).

| SD | TD | Pruned | STD Accuracy on FoVs(%) | | | Modularity(%) | | Compactness(%) | |
|---|---|---|---|---|---|---|---|---|---|
| | | | Category (51) | Elevation (4) | Pose (263) | Our (OS) | DCI | Our (MES) | MIG |
| dSprites | RGBD (depth) | ✗ | 0.01 (1.07) | 0.01 (0.89) | 0.00 (0.07) | 0.01 (0.47) | 0.02 (4.03) | 0.01 (0.54) | 0.01 (2.47) |
| | | ✓ | 0.07 (7.45) | 0.05 (5.32) | 0.00 (0.10) | | | | |
| Shapes3D | | ✗ | 0.01 (1.07) | 0.01 (0.89) | 0.00 (0.07) | 0.01 (0.54) | 0.02 (4.03) | 0.01 (0.54) | 0.01 (2.47) |
| | | ✓ | 0.07 (7.45) | 0.05 (5.32) | 0.00 (0.10) | | | | |
| Coil(bin) | | ✗ | 0.33 (38.97) | 0.17 (17.92) | 0.00 (0.07) | 0.01 (0.53) | 0.05 (5.73) | 0.01 (0.57) | 0.02 (1.31) |
| | | ✓ | 0.16 (20.37) | 0.12 (13.59) | 0.00 (0.09) | | | | |
| C-dSprites | RGBD | ✗ | 0.01 (0.58) | 0.02 (1.22) | 0.00 (0.06) | 0.00 (0.39) | 0.02 (2.39) | 0.01 (0.77) | 0.01 (0.85) |
| | | ✓ | 0.08 (13.95) | 0.03 (6.62) | 0.00 (0.07) | | | | |
| Coil | | ✗ | 0.01 (0.30) | 0.01 (0.87) | 0.00 (0.07) | 0.00 (0.36) | 0.01 (2.92) | 0.01 (0.67) | 0.01 (0.62) |
| | | ✓ | 0.04 (12.19) | 0.02 (5.64) | 0.00 (0.08) | | | | |
| Coil(bin) | | ✗ | 0.01 (0.32) | 0.02 (0.75) | 0.00 (0.09) | 0.00 (0.42) | 0.03 (2.43) | 0.01 (0.90) | 0.00 (0.64) |
| | | ✓ | 0.11 (13.39) | 0.05 (6.53) | 0.00 (0.09) | | | | |

