# OpenReview forum: "Transferring disentangled representations: bridging the gap between synthetic and real images"
_NeurIPS.cc/2024/Conference — NeurIPS 2024 poster_

### Official Review · Reviewer_Fxij · 2024-06-26

**Soundness:** 4
**Presentation:** 3
**Contribution:** 3
**Rating:** 7
**Confidence:** 3

**Summary:**

The work is situated in the area of Disentangled Representation Learning. The authors propose a new intervention-based metric (OMES) to assess the degree of disentanglement in different models. They perform extensive and comprehensive experiments validating and comparing OMES with other metrics. They thoroughly discuss the behavior of their metric with respect to other metrics and provide intuitions for the observed behavior. Finally, they use OMES to study transfer in disentangled representations from a source to a target dataset. The authors wish to understand which disentanglement properties remain after the transfer which they measure with OMES (and other metrics).

**Strengths:**

The paper is very well written and well structured. The OMES metric is explained and motivated well. The comparisons to the other metrics are comprehensive and sensible. I especially appreciate the formulation of the main research questions in Section 3.1 which guide the logic of the paper.

The conducted experiments are very extensive and I appreciate the additional results in Appendix C.

**Weaknesses:**

I do not see major weaknesses with this paper. I think this is a solid and a mature submission. I have some comments and questions which should be well addressable.

I found the results section in 3.3. quite hard to read. Each of the transfer experiments consists of one large block of text which is a bit hard to parse. I encourage the authors to think about restructuring it and introducing more paragraphs to promote readability.

Fig.1 left: the ylabel should be renamed to “Disentanglement score” for clarity.

Table 2: Please write a more descriptive caption


### Typos:
Line 33: tranferring

Line 169: Agreeement

Line 589: Table 7 not Fig 7

**Questions:**

OMES has one hyperparameter alpha in Eq. 1. As far as I have seen, it has always been set to 0.5. Does alpha need tuning / is 0.5 the optimal choice? What happens if alpha is set to a different value? Does this value depend on the dataset?

The authors assess the distance between the source and the target data in order to understand how this distance affects transfer performance. While the distance between two datasets is hard to measure, I felt the use of “distance” was a bit loose here, especially because the first research question explicitly involves the distance: “How well does disentanglement transfer, and how much does it depend on the distance between Source and Target Dataset?” In line 51, the authors write: “We discuss the role of fine-tuning and the need to reason on the distance between Source and Target datasets.” I have not really seen any reasoning on the distance between the Source and Target datasets in the paper and would be interested to hear how the authors think about it. The distance could be measured in FoV or in the pixel space or in the “domain” space, for example. We could have the datasets A, B and C where A has 5 FoVs and consists of real images. Then, B would have 4 FoVs (a subset of A) and consist of synthetic, but realistic looking images. Finally, C would have the same 5 FoVs as A, but would consist of sketches, that is, the image distribution would be very far away from A and B. Which dataset would then be closer to A in terms of “distance”: B or C? And how would we expect the disentanglement measures to transfer? I would think that the trained VAE heavily depends on the image domain and switching real images to sketches would destroy all performance in terms of disentanglement.

**Limitations:**

I think a paragraph which explicitly discusses the limitations of the approach would be beneficial for the paper.

---

> ### Author Rebuttal · Authors · 2024-08-06
>
> We thank the reviewer for considering our work and for their valuable comments and insights.
> * > I found the results section in 3.3 quite hard to read ... promote readability.
>
>    We agree with the reviewer that, mainly due to space limitations, the experimental section is very compact. We will attempt to re-design the section to improve its readability (for instance with bulleted lists if more space will be allowed) and will try to refer more explicitly to the research questions to provide clearer storytelling in our discussion.
>
> * > Fig.1 left: the label should be renamed ...  for clarity.
>
>    The reviewer is right, we will correct the label.
>
> * > Table 2: Please write a more descriptive caption
>
>    We agree with the reviewer that a more descriptive caption is needed for readability. Unfortunately, we sacrificed the captions (the one in Table 2, but also some others in the following) for space reasons. We will revise it in compatibility with space constraints. A possible caption is the following: “Disentanglement metrics of transfer to XXX (Dataset Target). Average scores before and after finetuning; in brackets, is the difference between finetuning and direct transfer.”
>
> * > Typos:
>
>    We thank the reviewer for pointing this out. We will correct all the typos in the revised version of the paper.
>
> * > OMES has one hyperparameter alpha ... on the dataset?
>
>    We agree with the reviewer that the text lacks an appropriate discussion on alpha. We will add some comments on the main paper, and we will provide more details in the Appendix, where an empirical analysis on the role of alpha was actually already provided (Appendix B.3), but without an appropriate and focused discussion.
>
>     Alpha does not need a specific tuning, since its role and behaviour are predictable by design. With reference to Eq. 1 we start observing that with alpha=0, OMES only measures the Compactness (MES) of the representation; with alpha=1, instead, our metric measures the Modularity (OS) only. Values of alpha in the interval (0,1) can be used to balance the importance of both contributions. In this sense, we can say it does not depend on the dataset, while it depends on which property(ies) we want to evaluate on the model. This is also implicitly shown in our experimental analysis (Sec. 2.3) where different datasets have been employed.
>
>     In Appendix B we report the results with different values of alpha, in which we show the behaviour of OMES when changing its value. In the main paper, we considered instead the general case of alpha=0.5 (*i.e.* compactness and modularity have the same importance) in the absence of valid reasons to favour one property or the other. We think this choice also allows us to have a fair comparison with the other disentanglement metrics.
>
> * > The authors assess the distance between the source and the target data... destroy all performance in terms of disentanglement.
>
>    Please refer to the common response.
>
> * > I think a paragraph which ... would be beneficial for the paper.
>
>    We agree with the reviewer that a section which more explicitly discusses the limitations of our approach would be beneficial for our work. In the present version of the paper, as observed by Rev. GNcy, we only discuss the limitations of transferring disentangled representations in Sec. 3.3, when commenting on the experimental analysis. Other important limitations of our current work are the use of a specific family of approaches (VAE-based, as noticed by Revs. GNcy and R67v) and the lack of a more considered strategy to reason on the distance between Source and Target datasets (this reviewer and Rev. GNcy), that may give insights on the best choice for the model to transfer (*i.e.* Source) depending on the specific task (*i.e.* Target) at hand.
>
>     We will add details on the main paper, if possible (given the space constraints), or alternatively in the Appendix.

---

> > ### Comment · Reviewer_Fxij · 2024-08-13
> > **Response to the rebuttal**
> >
> > I have read the rebuttal, the other reviews and the comments. I thank the authors for their responses and clarifications and I am keeping my original score.

---

### Official Review · Reviewer_R67v · 2024-07-02

**Soundness:** 3
**Presentation:** 2
**Contribution:** 2
**Rating:** 6
**Confidence:** 5

**Summary:**

This paper conducts an empirical investigation into the problem of transferring disentangled representations from synthetic data to real-world data (syn2syn, syn2real, real2real). They start with well-defined research questions and perform the investigation on the feasibility and effectiveness of transfer learning step by step. Besides the investigation, this paper also proposes an intervention-based metric measuring the quality of factor encoding in the representation while providing information about its structure.

**Strengths:**

1. The proposed metric OMES (Overlap Multiple Encoding Scores) is designed to measure two properties of disentangled representations: modularity and compactness. The proposed metric agrees well with other established metrics and also agrees well with performance metrics.

2. The experiments are extensive and comprehensive, covering four application scenarios (syn2syn, syn2real, real2real) and supported by six datasets. Some empirical insights are also revealed from the experiments. For example, the authors suggest that Explicitness is usually well maintained, while Modularity and Compactness are reduced as we move from synthetic to real. Interestingly, the authors also found out that fine-tuning is always beneficial, which is not an expected behavior to me.

**Weaknesses:**

1. The presentation of the proposed metric has poor readability and is not structured well. The description from line85 to line132 is a bit messed up: for example, the input image pair and the motivation for choosing subsets of latent dimensions at line102 should be described earlier, while it might be more appropriate to move the high-level description to the introduction. I would suggest the authors add a few headlines to the metric introduction at least or rephrase the contents.

2. Could you please consider synthetic datasets composed of more complex transformations, such as Falcol3D and Issac3D [1]? Though the datasets in the paper are very diverse, I feel a bit the transformations are relatively too simple. It is not sure to me if these insights transfer to real-world complex transformations. In particular, is fine-tuning still beneficial?

3. How about vector-based disentanglement methods [2][3]? How do the metrics and experiments generalize to vector-based methods? Do the authors think the insights will also transfer to these vector-based approaches?


> [1] Semi-supervised stylegan for disentanglement learning. ICML. 2020.
>
> [2] Flow Factorized Representation Learning. NeurIPS. 2023.
>
> [3] Unsupervised Learning of Disentangled Representations from Video. NeurIPS. 2017.

**Questions:**

Please see the weaknesses.

**Limitations:**

Please refer to the weaknesses. There are no particular limitations of this paper -- there is a common limitation of disentangled representation learning that these approaches are not scalable to large datasets.

---

> ### Author Rebuttal · Authors · 2024-08-06
>
> We thank the reviewer for the valuable insights provided with their comment. In the response, we will follow the structure of the bullet list in the review.
> * > The presentation of the proposed metric has poor readability... at least or rephrase the contents.
>
>    We agree with the reviewer that this section would benefit from a revision to improve the readability (which is probably not optimal also due to the space constraints). As suggested by the reviewer, we will anticipate the details on the setup and the main design choices, and we will move the high-level description and motivations to the introduction. We hope this will help the clarity and readability of this important section of our work.
>
> * > Could you please consider synthetic datasets composed of more complex transformations ... is fine-tuning still beneficial?
>
>      We thank the reviewer for the observation, which allows us to further elaborate on the strengths of our proposed methodologies.
>
>      Considering the limited time for the rebuttal, among the two datasets suggested by the reviewer, we opted for Isaac3 because we find it to be more in line with the scenarios already considered, while providing a considerably higher complexity, with its 9 latent factors.
>
>      With Isaac3D, we performed some further experiments by using the new dataset both as a Source and as a Target, and following the procedure described in Sec. 3.1 and common to all the experiments already included in the paper. The results of these new experiments are reported in the pdf attached to the rebuttal. More specifically, in Tables 1 and 2 we report the results when Isaac3D is used as a Target. In this case, the models trained on Shape3D have been used as a Source.
>
>      We observe that the transfer is overall effective according to all the disentanglement scores. If we focus in particular on the “All” column, we notice that the performance here is higher than the one we obtain when Shape3D is Source and Coil is Target (Tab. 4 in the main paper) or RGBD is Target (Tab. 5 in the main paper). This is in line with the “picture” provided by OMES (Isaac3D as Target: 23,15, which becomes 33.1 after finetuning; Coil as Target: 28.35, and 27.95; RGBD: 28.85 and 28.2). We notice that only for Isaac3D the boost of finetuning is appreciable, while the baseline result is lower than for Coil and RGBD (that are real datasets). Overall, these empirical evidences may suggest that the robustness of transferring disentangled representations is influenced by several factors including the Sim2Real challenge but also the number of factors, their granularity, and how many factors Source and Target have in common.
>
>      This is related to a more general concept of distance between datasets, we will consider in our future investigations and that we briefly discuss in the general response.
>
>      On the other hand, Table 3 of the attached pdf reports the transfer from Isaac3D as the Source to Shape3D as the Target. Although the richer transformations of Isaac3D, the model adaptation with finetuning is still beneficial for all the disentanglement metrics. We think this may be explained by the “domain” dependence of  VAE models, for which an adaptation step is in general beneficial.
>
> * > How about vector-based disentanglement methods ... insights will also transfer to these vector-based approaches?
>
>      Please refer to the common response.

---

> > ### Comment · Reviewer_R67v · 2024-08-11
> > **Thanks for the response**
> >
> > Thanks for the response!
> >
> > I like the new experiments of Isaac3D and I will keep my original score.

---

### Official Review · Reviewer_GNcy · 2024-07-11

**Soundness:** 3
**Presentation:** 3
**Contribution:** 2
**Rating:** 6
**Confidence:** 3

**Summary:**

This paper proposes a novel classifier-free metric for quantitatively measuring disentanglement and investigates transferring disentangled representations from a synthetic dataset with ground truth factors of variation (FoV) to a target real-world dataset. The authors introduce OMES, a novel intervention-based metric that evaluates the correlation between representations of two images sharing all FoVs. OMES computes the overlap between different FoVs (Overlap Score) and measures the encoding of each factor within single dimensions (Multiple Encoding Score). Using these metrics, the paper analyzes the properties of disentangled representations and source/target distributions to improve disentanglement transfer. The main contributions of the paper are:
- Introduction of a novel classifier-free metric (OMES) for disentanglement evaluation, which reduces hyper-parameter sensitivity.
- Extensive empirical study on transferring disentangled representations from source to target datasets, revealing the potential and properties of disentanglement transfer.

**Strengths:**

- The paper is easy to follow.
- A simple yet novel classifier-free metric removes the dependency on hyper-parameters and enables reasonable comparison between various configurations and benchmarks. Also, this metric maintains the reasonable assessment of disentanglement compared to the conventional metrics and provides an informative tool for analyzing the each factor of variations.
- Thorough empirical analysis provides comprehensive understanding on a novel metric and disentanglement transfer learning.

**Weaknesses:**

- The implications on transferring disentanglement from synthetic datasets to complex real-world datasets are somewhat limited. Although the empirical study indicates that a smaller distance between the source and target datasets and shared FoVs between them are beneficial, these are expected properties of conventional transfer learning. The paper would be strengthened by discussing how to specifically select a proper source dataset for a given target dataset with unknown factors of variation. For example, it would be useful to define and measure the structural similarity between a target dataset and potential source datasets.
- The presentation could be improved. At first glance, it feels like two independent topics (metrics and transfer learning) are being introduced, making it hard to understand why a novel metric is needed for transferring disentangled representations. It would help readers if the connections between these two components were more clearly explained.

**Questions:**

- In transferring disentangled representation learning, how much of the improvement comes from transfer learning compared to training from scratch on the target dataset, in terms of disentanglement metrics? This comparison would provide a clearer understanding of the benefits and effectiveness of the transfer learning approach.
- Can you provide more analysis on why OMES has relatively low correlation on FactorVAE score in Figure 2?
- Most of the investigations are done on VAE-based models. Would it have the same implications with other recent disentangled representation learning approaches [1, 2, 3], which employ more powerful generative models?


[1] Yang, et al. "Disdiff: Unsupervised disentanglement of diffusion probabilistic models.", in NeurIPS 23.

[2] Lin et al., “Infogan-cr and modelcentrality: Self-supervised model training and selection for disentangling gans”, In ICML 20.

[3] Ren et al., “Learning disentangled representation by exploiting pretrained generative models: A contrastive learning view”, in ICLR 21.

**Limitations:**

- Section 3.3 describes the limitation of transferring disentanglement learning.

---

> ### Author Rebuttal · Authors · 2024-08-06
>
> We thank the reviewer for their effort spent in providing valuable feedback on our work. In the response, we will follow the paragraph listing structure of the review.
>
> * > The implications on transferring disentanglement from synthetic datasets ... and potential source datasets.
>
>     Please refer to the common response.
>
> * > The presentation could be improved...were more clearly explained.
>
>      We agree with the reviewer that the paper is rather dense in some parts, and this does not help the understanding. We will revise the contributions of our work to better clarify the connections between them in the final version of the paper, should it be accepted.
>
>      The novel metric is not strictly required by the application to transfer learning, but rather by more general needs for disentanglement metrics, the first being interpretability. We also wanted our metric not to be based on a classifier and to be able to estimate different properties of disentangled representations (*i.e.* modularity, compactness) simultaneously.
>
>      In this sense, the contributions of our work are indeed two, somehow independent: the new metric OMES and the transfer learning methodology. However, it is worth noticing that the interpretability of OMES allows us to directly exploit it in the transfer learning process, for instance, to select the most representative dimension of the representation for the classification experiments (we named “Pruned” in all our tables).
>
> * > In transferring disentangled representation learning, how much ... effectiveness of the transfer learning approach.
>
>      We thank the reviewer for the question that allows us to illustrate a different aspect of our approach. We start observing that training from scratch on the Target dataset is usually not possible in the application domain we have in mind, where the annotation of the latent factors is not available in most cases. This explains why we did not consider this scenario in our experimental analysis. Nevertheless, we performed a further evaluation to quantify the loss of transferring rather than training from scratch.
>
>      We used the source models trained directly on Shapes3D, Coil100 and Isaac3d (which we trained under the suggestion of Rev. R67v)  and we compared their performances with the models trained on different Sources and finetuned on the 3 datasets.  The results are reported in Table 4 of the attached pdf. It reports the average scores of the models trained from scratch (with weak supervision).  In brackets instead, we put the average differences between the scores obtained from transfer learning + finetuning, and the ones with training from scratch (*i.e.* a negative value means we are losing when applying the transfer methodology).
>
>      In general, we observe that the models trained from scratch perform almost always better than the transferred ones. This gap is expected because the models trained from scratch use the annotation of the dataset, while the finetuning is unsupervised. However, in this experiment, the gap is noticeable only when Shape3D is a Target, this could be explained by the fact that it has relatively simple transformations easier to disentangle when the annotation of factors is used for training. On the contrary, for datasets with real-world more complex (harder to learn and disentangle) transformations,the performances with finetuning is closer to the training from scratch. We also observe that, given the complexity, their overall performance is lower than the ones of Shape3D.
>
>
> * > Can you provide more analysis on why OMES ... FactorVAE score in Figure 2?
>
>     We first observe that even if they are all intervention-based metrics, the formulation of the input pairs of our metric and FactorVAE/BetaVAE are the opposite (we sample pairs of images where all factors are the same but one, while FactorVAE and BetaVAE fix one common factor while randomly sampling the values of the others). The correlation of OMES with FactorVAE is poor (28) but an even lower correlation can be observed with BetaVAE (11). The latter can be explained by observing that when computing OMES we consider contributions from (latent dimension, factor) associations accurately selected to maximize the correlation, while  BetaVAE keeps the information from the entire latent representations (despite some dimensions might be uninformative or even potentially harmful). Concerning FactorVAE instead, we notice that it essentially estimates the performance of a linear classifier on input-output pairs composed of the fixed latent factor (output) and the latent dimension which presumably mostly encodes the factor (input). This step is also a core for OMES, although implemented following a different strategy (*i.e.* we rely on high correlation rather than low variance to select the “best” encoding factor).
>
>     Among the properties that may explain the different behaviours of OMES and FactorVAE/BetaVAE in Fig. 2 there is also the fact they measure only the modularity of disentanglement, while our metric OMES also accounts for compactness since alpha=0.5 (this comparison is reported in Table 6 in the Appendix). (For more details on the role of alpha, we refer the reviewer to the answer to Rev. Fxij on the same topic).
>
>
> * > Most of the investigations are done on VAE-based models ... employ more powerful generative models?
>
>    Please refer to the common response.

---

> > ### Comment · Reviewer_GNcy · 2024-08-11
> > **Response to Author Rebuttal**
> >
> > I appreciate the authors' effort in addressing the concerns.
> > Most of the concerns are clarified. Although I still believe that including experiments on other disentangled representation learning methods would provide a more comprehensive understanding, I agree that the authors' transferring method is not limited to specific methods and this is not a crucial concern.
> >
> > However, my concern about the comparison to a model trained from scratch remains unresolved.
> > If I understand correctly, their experiments involve two steps: (1) weakly supervised learning (Ada-GVAE) on the Source Dataset, and (2) fine-tuning the model in an unsupervised manner (which does not require GT FoVs). I was asking for the performance gap between step (1) -> (2) (the proposed method) and step (2) alone (training from scratch on the target dataset) to evaluate the impact of step (1). However, it seems that the authors reported the performance of weakly supervised learning (Ada-GVAE) on the Target Dataset, which does not clarify the effect of transferring from the Source Dataset. Please correct me if I have misunderstanding here.

---

> > > ### Author Response · Authors · 2024-08-12
> > >
> > > We thank the reviewer for appreciating the rebuttal.
> > >
> > > We now realise we misunderstood one request. Indeed, in Table 4 of the pdf we attached to the rebuttal we reported the performances of the weakly-supervised model trained on the Target dataset and the differences (in brackets) of the scores obtained on the Target in this way or with our full pipeline, while the reviewer was asking for a comparison with the unsupervised approach directly applied to the "target" dataset.
> > >
> > > In the earliest stages of our work, we actually adopted the unsupervised approach on our reference datasets and we empirically observed its limitations emerging already on “simple” datasets. For this reason, we opted for some level of supervision.
> > >
> > >  To give a quantification of what we gain with this change, we report here the score obtained in our original unsupervised experiments focused on the use of two synthetic datasets, namely Color-dSprites and Noisy-dSprites (for details on the dataset see the main paper). With unsupervised training from scratch on Color-dSprite we have DCI=0.14 and MIG=0.07, which become respectively 53.3 and 34.3 with our full pipeline based on transfer learning (Source dataset: dSprites).
> > >
> > > On Noisy-dSprites, with unsupervised learning from scratch, we obtained DCI=0.05 and MIG=0.02, which become 26.0 and 36.0 respectively with our full approach.
> > > As a reference, training from scratch on the two datasets using Ada-gVAE we have DCI=64.2 and MIG=49.5, showing the large expected gap between unsupervised and weakly-supervised training from scratch.

---

> > > > ### Comment · Reviewer_GNcy · 2024-08-13
> > > > **Response to the Authors' comment**
> > > >
> > > > Thank you for the detailed comment. I think the improvements seems significant (I guess the scales of DCI and MIG are just typo? e.g., DCI=0.14 means 14.0 I guess?) and the impact of transferring disentangled representation from the source dataset is now clear. Now I have no remaining concerns so I will raise the score to weak accept. I thank the authors for the detailed response.

---

### Author Rebuttal · Authors · 2024-08-06

We thank the reviewers for their efforts in reading our paper and also for providing valuable insights and new interpretations for our work. There is a general agreement on the effectiveness of the new metric and the extensive assessment with a thorough experimental analysis.

In this common response, we address comments shared by more than one reviewer. We are responding to each reviewer independently on the remaining points.
* Rev. **GNcy** and Rev. **Fxij** express concern about *how to assess the distance between the Source and the Target datasets*. While a discussion on datasets distance is out of the scope of this paper, the reviewers' comments lead us to see a potential application of our approach in future research.

  It is true the word “distance” may be not appropriate, in a sense we are assessing *how two datasets are related mainly in terms of FoVs*:
  1. If both Source and  Target are synthetic, we normally possess a prior on their FoVs and we may reason on their similarity in a structured manner (choosing Source and Target with a majority of identical factors, such as dSprite and color-dSprite, or with similar factors such as Coil100 and BinaryCoil100).
   2. If the Target is real, we may not possess a proper FoVs annotation but we may qualitatively identify some dominant FoVs of interest, which we could use to find an appropriate Source where such factors are present.
   3. We notice that, even in the supervised case, FoVs are often similar but not identical, they may differ in internal variability/granularity (*e.g.* *Scale* factor in dSprites and Shapes3D, *Pose* in Coil100 and RGBD-Objects), or may have the same name but be different in nature (*e.g.* *Orientation* in dSprites and Shapes3D).

   We conducted the transfer experiments considering different scenarios (syn2syn, syn2real and real2real) to verify which disentanglement properties are preserved or degraded with the transfer. For this, we reasoned on scenarios of increasing complexity, including, for instance, scenarios where the Target has the same FoVs of the Source plus an additional one, or where the Source is a “simplified” version of the Target (*e.g.* from a binarized version of Coil100 to Coil100).  Interestingly, as reported in Sec. 3.3 using a real dataset as a Source does not provide particular improvement in performance with respect to the use of a synthetic Source if we apply the fine-tuning.

   In the case of real data, this might suggest that *using a synthetic Source dataset* (well related to the Target one in terms of FoVs) and adapting the model with finetuning can (in principle) *provide similar performances to models trained on real images*, and hence closer in pixel and domain space. In the revised version of the paper, we will add a note on the choice of the Source depending on the Target.

***

* Rev. **GNcy** and Rev. **R67v** observe that our methodology mostly employs VAE-based methods and *ask about the implications of using more recent/powerful methods*. We agree with the reviewers that an analysis of the generalisation of our
insights to different disentanglement learning approaches is needed, and it will be the object of our investigations in the near future.

   Our decision to consider VAE-based models in this work, and Ada-GVAE in particular, is due to their *simplicity*. Keeping the complexity of the disentanglement learning approach under control, we are relatively confident that the benefits we might observe are due to the transfer methodology of disentanglement rather than to the high expressive power of complex models for disentangled representation learning. It is also worth mentioning that the sampling strategy of Ada-GVAE methods is similar to the one we adopt for OMES, and this allows us to obtain an overall coherent procedure.

   To add some more observations on the approaches suggested by the reviewers, we observe that recent works based on generative mechanisms belong to the vector-based family of approaches. This means that they adopt more than one dimension in the form of a vector to encode a generative factor, while dimension-based methods (*e.g.* VAE) usually adopt a single dimension to represent the factor. For this reason, vector-based methods are more suitable for coarse-grained factors (capturing more information) while dimension-based methods are suitable for fine-grained factors. As observed in [A], the latter is more appropriate when investigating under-explored research directions (as in our case) being more precisely defined.


   All the existing disentanglement metrics discussed in the work have been proposed for evaluating VAE-based methods, including ours. However, they can be easily used to evaluate vector-based methods, for example, performing PCA as post-processing on the representation before evaluation, as already done in the literature [B, C].
   We have no particular reasons to think our insights are not transferring to vector-based approaches. However, each family of approaches for disentanglement learning *follows specific paradigms that may require tailored designs for transfer learning*. In other words, while the general transfer methodology is still applicable, it may need proper tuning to perform optimally depending on the particular learning approach.

   The generalization of our methodology to different disentanglement learning approaches will be object of our future investigation, since they represent a further step forward from the current state of our work. We will include the suggested works in the bibliography of this paper, referring to them in conclusions and future works.


   [A] Wang, et al. "*Disentangled representation learning.*" arXiv:2211.11695; PAMI 2024 (accepted)

   [B] Yang, et al. "*Disdiff: Unsupervised disentanglement of diffusion probabilistic models.*", in NeurIPS 23.

   [C] Du, et al. “*Unsupervised Learning of Compositional Energy Concepts*” in NeurIPS 21.

---

### Decision · Program_Chairs · 2024-09-25

**Decision:**

Accept (poster)

**Comment:**

The authors make a solid contribution to the field of disentangled representation learning by introducing OMES, a novel metric for evaluating disentanglement, and by providing a comprehensive empirical study on disentanglement transfer. While the paper is technically sound and the proposed metric has the potential for significant impact, there are areas for improvement, particularly in the presentation and generalization of the findings. Please use the provided feedback to enhance clarity and discuss broader applicability of the results.